# Non-Uniform Multiclass Learning with Bandit Feedback

**Steve Hanneke**
Department of Computer Science
Purdue University
West Lafayette, IN 47907
`steve.hanneke@gmail.com`

**Amirreza Shaeri**
Department of Computer Science
Purdue University
West Lafayette, IN 47907
`amirreza.shaeiri@gmail.com`

**Hongao Wang**
Department of Computer Science
Purdue University
West Lafayette, IN 47907
`wang5270@purdue.edu`

## Abstract

We study the problem of multiclass learning with bandit feedback in both the i.i.d. batch and adversarial online models. In the *uniform* learning framework, it is well known that no hypothesis class $\mathcal{H}$ is learnable in either model when the effective number of labels is unbounded. In contrast, within the *universal* learning framework, recent works by Hanneke et al. [2025b] and Hanneke et al. [2025a] have established surprising exact equivalences between learnability under bandit feedback and full supervision in both the i.i.d. batch and adversarial online models, respectively. This raises a natural question: What happens in the *non-uniform* learning framework, which lies between the uniform and universal learning frameworks? Our contributions are twofold: (1) We provide a combinatorial characterization of learnable hypothesis classes in both models, in the realizable and agnostic settings, within the non-uniform learning framework. Notably, this includes elementary and natural hypothesis classes, such as a countably infinite collection of constant functions over some domain that is learnable in both models. (2) We construct a hypothesis class that is non-uniformly learnable under full supervision in the adversarial online model (and thus also in the i.i.d. batch model), but not non-uniformly learnable under bandit feedback in the i.i.d. batch model (and thus also not in the adversarial online model). This serves as our main novel technical contribution that reveals a fundamental distinction between the non-uniform and universal learning frameworks.

## 1 Introduction

Learnability lies at the core of learning theory. Intuitively, it explores the conditions under which there exists an algorithm that can accurately predict unseen instances based on a finite set of examples. To rigorously study this phenomenon, various abstract learning frameworks, including uniform, non-uniform, and universal learning frameworks, have been proposed in the literature so far. This manuscript focuses on the non-uniform learning framework.

The classical work of Benedek and Itai [1988] introduced the non-uniform learning framework. Moreover, they gave a combinatorial characterization of non-uniform PAC learnability for the binary classification setting. The recent contribution of Lu [2024] gave a combinatorial characterization of

39th Conference on Neural Information Processing Systems (NeurIPS 2025).

non-uniform online learnability also for the binary classification setting. However, many important tasks involve a large prediction space. For instance, consider a quantum experiment with $n \in \mathbb{N}$ qubits, where the goal is to predict measurement outcomes, each of which can be one of $2^n$ possible states. Meanwhile, many significant tasks involve bandit feedback, where the only information that can be received is whether the prediction is correct. For example, a physicist might propose a prediction based on a discovered law of nature and then observe only whether it holds true.

In response, we consider the framework called "Non-Uniform Multiclass Learning under Bandit Feedback". Roughly speaking, in this framework, a learner aims to infer an unknown hypothesis $h^\star$ from a known hypothesis class $\mathcal{H}$ consisting of functions from some instance space $\mathcal{X}$ (e.g., space of images) to some label space $\mathcal{Y}$ (e.g., categories of images), through an interaction with nature, with a guarantee that may depend on $h^\star$.

Subsequently, we provide a more precise, yet still informal, description of the framework. In the i.i.d. batch model, nature initially selects an unknown data distribution similar to classical multiclass learning, but the learner does not directly observe the correct labels of the i.i.d. training examples. Instead, during each round, the learner first receives an unlabeled example, makes a prediction for its label, and receives bandit feedback. Despite this restriction, the goal remains the same as in classical multiclass PAC learning, where the primary objective is to output a function that correctly classifies most future examples generated by the same underlying data distribution. In addition, in the adversarial online model, during each round, nature first presents an instance to the learner. Next, the learner must predict a label for the given instance. After making a prediction, nature reveals the bandit feedback. Crucially, this setting differs from the full supervision setting, where, at the end of each round, the true label is revealed by nature. Again, despite this restriction, the goal remains the same as in classical multiclass online learning, where the primary objective is to correctly predict in most rounds. Furthermore, in the realizable setting, we assume that the data distribution or the sequence of instance-label pairs played by the adversary is consistent with some $h^\star \in \mathcal{H}$. On the other hand, in the agnostic setting, we do not make such an assumption. Moreover, the non-uniform learning framework allows for error or regret bounds, possibly depending on $h^\star$, in the i.i.d. batch and adversarial online models, respectively.

In this manuscript, our first main contribution is to algorithmically address the following question in the non-uniform multiclass learning under bandit feedback framework across both the i.i.d. batch and adversarial online models, in both the realizable and agnostic settings:

> *What is the necessary and sufficient condition for a hypothesis class $\mathcal{H} \subseteq \mathcal{Y}^{\mathcal{X}}$ that admits non-uniform learnability?*

In particular, we provide a combinatorial characterization of hypothesis classes for which learnability is possible. We note that, for simplicity at this stage of the introduction, several details about the definitions in the framework have been omitted, such as the possibility of allowing randomized learners in the adversarial online model. For more details, see Section 1.1 and Section 3.

Previous research on the problem of multiclass learning under bandit feedback mostly considered the *uniform* learning framework. In essence, the uniform learning framework seeks a theoretical guarantee that is true for all data distributions or all adversarial sequences, without any dependence on the distribution or sequence itself. This line of study was initiated by the seminal work of Daniely et al. [2011] and continued by Erez et al. [2024b,a] in the PAC model, and by Daniely and Helbertal [2013], Long [2017], Geneson [2021], Raman et al. [2023], Hanneke and Yang [2023] in the adversarial online model. If the effective label space is infinite—that is, there exists at least one instance in the instance space for which hypotheses in the hypothesis class can assign infinitely many distinct labels—no deterministic online learning algorithm can guarantee a *uniformly* bounded number of mistakes against the worst-case realizable adversary. To see this, consider a scenario in which the adversary repeatedly selects the same specific instance in every round. In this case, the predictions of the online learning algorithm can be consistently incorrect, even though a hypothesis within the hypothesis class remains consistent with all previous feedback. In particular, this happens because the feedback received only indicates whether the prediction was correct or not. Moreover, this conclusion can be easily generalized to randomized algorithms as well. In addition, a similar limitation arises in the PAC model due to the need to guess an unknown natural number while keeping the number of guesses uniformly bounded over the choices of the target number. Notably, this limitation applies even to elementary and natural hypothesis classes, such as a countably infinite collection of constant functions over some domain.

The recent contributions by Hanneke et al. [2025b] and Hanneke et al. [2025a] are the only works that diverge from the uniform learning framework by considering the universal learning framework of Bousquet et al. [2021]. This framework enables the study of theoretical guarantees that are still true for all data distributions or all adversarial sequences, but crucially *without placing a uniform bound* on the error rate in the i.i.d. batch model or regret in the adversarial online model. In particular, the error rate may depend on the distribution itself, and the number of mistakes may depend on the infinite sequence of instance-label pairs. These works demonstrated surprising exact equivalences between learnability under bandit feedback and full supervision in the i.i.d. batch and adversarial online models, respectively. As a result, it is possible to learn the hypothesis class of a countably infinite collection of constant functions over some domain in the universal multiclass learning under bandit feedback framework in both i.i.d. batch and adversarial online models as opposed to the uniform learning framework.

This raises a natural question: what happens in the *non-uniform* learning framework, which lies between the uniform and universal learning frameworks? In particular, in the former, learnability is essentially impossible under bandit feedback, whereas in the latter, there is an exact equivalence between learnability under full supervision and learnability under bandit feedback.

The non-uniform learning framework also allows one to escape the aforementioned theoretical limitation in the uniform learning framework. To illustrate this, we again focus on the hypothesis class of countably infinite constant functions over some domain. Suppose that we are in the adversarial online model and in the realizable setting. Now, consider the following online learning algorithm: In each round $i \in \mathbb{N}$, it predicts the label $i$ until it receives feedback confirming the prediction is correct. Once the correct label is identified, it consistently predicts that label in all subsequent rounds. This online learning algorithm makes a finite number of mistakes, depending on $h^\star$, against any realizable adversary. It is not hard to see that this class is non-uniform online learnable in the agnostic setting as well as non-uniform PAC learnable in both realizable and agnostic settings. However, we construct a hypothesis class that is non-uniformly learnable under full supervision in the adversarial online model (and thus also in the i.i.d. batch model), but not non-uniformly learnable under bandit feedback in the i.i.d. batch model (and thus also not in the adversarial online model). This serves as our main novel technical contribution that reveals a fundamental distinction between the non-uniform and universal learning frameworks.

To summarize, our main contributions in the current manuscript are as follows. Therefore, we complete the picture of the problem of multiclass learning, which is presented in the following table.

| | Adversarial | Adversarial with Bandit Feedback | I.I.D. | I.I.D. with Bandit Feedback |
|---|---|---|---|---|
| Uniform | Littlestone classes [Daniely et al., 2011] | bandit Littlestone classes [Daniely et al., 2011] | finite DS Dimension [Brukhim et al., 2022] | finite effecive label space and Natarajan dimension[Daniely et al., 2011] |
| Non-Uniform | countable union of Littlestone classes [Theorem D.1] | countable union of bandit Littlestone classes [Theorem 1.1] | countable union of DS classes [Theorem C.3] | countable union of classes with finite effective label space and Natarajan dimension [Theorem 1.3] |
| Universal | no infinite Littlestone Tree [Hanneke et al., 2025a] | no infinite Littlestone Tree [Hanneke et al., 2025a] | All [Hanneke et al., 2021] | All [Blanchard et al., 2022] |

Table 1: The Learnable classes in different multiclass learning settings.

We also provide several interesting examples that show the separations among different learning settings:

1. We construct a hypothesis class that is non-uniformly learnable under full supervision in the adversarial online model (and thus also in the i.i.d. batch model), but not non-uniformly

learnable under bandit feedback in the PAC model (and thus also not in the adversarial online model), in Proposition B.1.

2. We give two examples of hypothesis classes to show other separations between uniform, non-uniform, and universal learning frameworks for the problems of multiclass learning under bandit feedback and full supervision Appendix B.

## 1.1 Overview of the Main Results

In the following subsection, we provide a detailed summary of the key results and findings presented in our paper.

### 1.1.1 Non-Uniform Multiclass Online Learning

We define the problem as a repeated game between the learner and the adversary. At each round $t \in \mathbb{N}$, the adversary selects an instance $X_t$ from an arbitrarily non-empty instance space $\mathcal{X}$ and a label $Y_t$ from a possibly countably infinite non-empty label space $\mathcal{Y}$ [1]; then, reveals $X_t$ to the learner. Subsequently, the learner predicts a (potentially randomized) label $\hat{Y}_t$ from $\mathcal{Y}$. Following this, the learner receives feedback. If we are in the full supervision setting, the feedback is $Y_t$; and if we are in the bandit feedback setting, the feedback is $\mathbb{I}\{\hat{Y}_t \neq Y_t\}$, only indicating whether the prediction is correct. Following standard learning theory conventions, we define a hypothesis class $\mathcal{H}$ as a set of functions from $\mathcal{X}$ to $\mathcal{Y}$. This class is known to the learner before the game begins. For additional details, see Section 3.

In the realizable setting, we assume that the sequence $\{(X_t, Y_t)\}_{t=1}^{\infty}$, chosen by the adversary, is consistent with at least one hypothesis in $\mathcal{H}$. In this setting, we focus on the standard notion of the expected number of mistakes made by the learner over time. We say that a hypothesis class $\mathcal{H} \subseteq \mathcal{Y}^{\mathcal{X}}$ is realizable non-uniform multiclass online learnable under bandit feedback if there exists an online learning rule $\mathbf{A}$ receiving bandit feedback such that for every $h^\star \in \mathcal{H}$ there exists a constant $c \in \mathbb{N}$ such that for every sequence $\{(X_t, Y_t)\}_{t=1}^{\infty}$ consistent with $h^\star$, the online learning algorithm $\mathbf{A}$ only makes $c$ number of mistakes in expectation. We have a similar definition for the full supervision setting. See Section 3 for further details. The main result of this part is the following theorem.

**Theorem 1.1.** *Let $\mathcal{H} \subseteq \mathcal{Y}^{\mathcal{X}}$ be a hypothesis class. Then, the following statements are equivalent.*

- *$\mathcal{H}$ can be represented as a countable union of hypothesis classes with finite bandit Littlestone dimension.*

- *$\mathcal{H}$ is realizable non-uniform multiclass online learnable under bandit feedback.*

To prove the upper bound of the above theorem, we design a novel online learning algorithm called the non-uniform bandit standard optimal algorithm. Furthermore, the proof of the lower bound is based on the original work of Benedek and Itai [1988]. As an immediate implication, consider the hypothesis class of a countably infinite collection of constant functions over some domain. Indeed, one can view each hypothesis in this class as part of the countable union of hypothesis classes with finite bandit Littlestone dimension. Thus, this hypothesis class is realizable non-uniform multiclass online learnable under bandit feedback. See Section 5 for further details. Notably, we also prove a similar theorem for the full supervision setting. The proof of this theorem is based on the arguments of the recent work of Lu [2024]. See Appendix D for further details.

In the agnostic setting, we make no assumptions about the sequence $\{(X_t, Y_t)\}_{t=1}^{\infty}$, chosen by the adversary. In this setting, our focus shifts to minimizing the standard notion of expected regret, which compares the expected number of mistakes made by the learner to those made by the best hypothesis in the hypothesis class $\mathcal{H}$ over the sequence. We say that the hypothesis class $\mathcal{H} \subseteq \mathcal{Y}^{\mathcal{X}}$ is agnostic non-uniform multiclass online learnable under bandit feedback if there exists an online learning rule $\mathbf{A}$ receiving bandit feedback such that for every $h^\star \in \mathcal{H}$ there exists a constant $c \in \mathbb{N}$ such that for every $T \in \mathbb{N}$ and every sequence $\{(X_t, Y_t)\}_{t=1}^{T}$, the online learning algorithm $\mathbf{A}$ has $o(c\,T)$ expected regret. We have a similar definition for the full supervision setting. See Section 3 for further details. The main result of this part is the following theorem.

---

[1] If we consider randomized learning algorithms, the associated $\sigma$-algebra is of little consequence, except that singleton sets $\{y\}$ should be measurable.

**Theorem 1.2.** *Let $\mathcal{H} \subseteq \mathcal{Y}^{\mathcal{X}}$ be a hypothesis class. Then, the following statements are equivalent.*

- *$\mathcal{H}$ can be represented as a countable union of hypothesis classes with finite bandit Littlestone dimension.*

- *$\mathcal{H}$ is agnostic non-uniform multiclass online learnable under bandit feedback.*

To establish the upper bound of the above theorem, we comprise several components. First, we prove a structural lemma for bandit Littlestone classes. Second, we adopt the technique from the original work of Benedek and Itai [1988]. Third, we use an ingredient from the recent work of Raman et al. [2023]. Finally, we employ the standard doubling trick. As an immediate implication, again consider the hypothesis class of a countably infinite collection of constant functions over some domain. Based on a similar argument as before, this hypothesis class is agnostic non-uniform multiclass online learnable under bandit feedback. See Appendix A.1 for further details, including the exact rate, which is $\tilde{\mathcal{O}}(\sqrt{T})$. Notably, we also prove a similar theorem for the full supervision setting. The proof of this theorem is similar to the proof of Theorem 1.2. See Appendix D for further details.

### 1.1.2 Non-Uniform Multiclass PAC Learning

Here, initially, nature selects an unknown data distribution $\mathcal{D}$ over $\mathcal{X} \times \mathcal{Y}$. Fix a sample size $n \in \mathbb{N}$. Subsequently, the learner and nature interact sequentially in $n$ rounds. In particular, during each round $t \in \{1, 2, \ldots, n\}$, nature first independently samples an example $(x_t, y_t) \sim \mathcal{D}$ and reveals $x_t$ to the learner. The learner is then tasked with predicting $\hat{y}_t \in \mathcal{Y}$ for the received instance. Upon the learner's prediction, nature only reveals whether the prediction is correct. Eventually, the learner must output a function from instance space to label space that correctly classifies most future examples generated by the same unknown data distribution. Roughly speaking this is called PAC learnability. Again, following standard learning theory conventions, we consider a hypothesis class $\mathcal{H}$, consisting of functions mapping $\mathcal{X}$ to $\mathcal{Y}$. We assume standard measurability assumptions on $\mathcal{X}$ and $\mathcal{C}$ as well. We say that a data distribution $\mathcal{D}$ is realizable by a hypothesis class $\mathcal{H}$ if there exists $h^* \in \mathcal{H}$ such that the samples are consistent with it almost surely. For additional details, see Section 3.

**Theorem 1.3.** *Let $\mathcal{H} \subseteq \mathcal{Y}^{\mathcal{X}}$ be a hypothesis class. Then, the following statements are equivalent.*

- *$\mathcal{H}$ can be represented as a countable union of hypothesis classes with finite effective label space and finite Natarajan dimension.*

- *$\mathcal{H}$ is realizable non-uniform multiclass PAC learnable under bandit feedback.*

- *$\mathcal{H}$ is agnostic non-uniform multiclass PAC learnable under bandit feedback.*

To prove the above theorem, we generalize the argument of the work of Benedek and Itai [1988] for the binary classification setting under full supervision using the results of Daniely et al. [2011] on PAC learnability under bandit feedback. As an immediate implication, consider the hypothesis class of a countably infinite collection of constant functions over some domain. Indeed, one can view each hypothesis in this class as part of the countable union of hypothesis classes with finite effective label space[2] and finite Natarajan dimension. Thus, this hypothesis class is realizable non-uniform multiclass online learnable under bandit feedback. See Section 4 for further details. Notably, we also prove a similar theorem for the full supervision setting. See Appendix C for further details.

### 1.1.3 Examples

We present three examples of hypothesis classes that demonstrate the separation between related learnability notions. Formally, we have the following results.

**Proposition 1.4.** *There exists a hypothesis class $\mathcal{H} \subseteq \mathcal{Y}^{\mathcal{X}}$ that is uniform multiclass online learnable under full supervision, but not non-uniform multiclass PAC learnable under bandit feedback.*

The above proposition reveals a fundamental distinction between the non-uniform and universal learning frameworks, where in later we have surprising exact equivalences between multiclass learnability under bandit feedback and full supervision. Indeed, if a hypothesis class is uniform

---

[2]The effective label sapce is the label space of one instance, that is, $\mathcal{Y}_x = \{h(x) : h \in \mathcal{H}\}$. We say a concept class $\mathcal{H}$ has finite effective label space, if and only if $\mathcal{Y}_x$ is finite for every $x \in \mathcal{X}$.

multiclass online learnable under full supervision, it is also uniform multiclass PAC learnable under full supervision, non-uniform multiclass online learnable under full supervision, and non-uniform multiclass PAC learnable under full supervision. In addition, if a hypothesis class is not non-uniform multiclass PAC learnable under bandit feedback, it is not also non-uniform multiclass online learnable under bandit feedback, uniform multiclass PAC learnable under bandit feedback, and uniform multiclass online learnable under bandit feedback. As a result, our theorem is stated in the most extreme case.

This serves as our main novel technical contribution. To prove it, the idea is as follows: Let $\mathcal{X}$ be the set of all non-empty countable sequences of distinct real numbers. Also, let $\mathcal{Y} = \mathbb{N} \cup \{\star\}$. In addition, let $\mathcal{H} = \{h_y \mid h_y : \mathcal{X} \rightarrow \mathcal{Y}, y \in \mathbb{R}, \forall_{S \in \mathcal{X}} \ h_y(S) = \star$ if $y$ isn't in the sequence $S$; otherwise $h_y(S) = i$, where $S[i] = y\}$. The crucial observation that we prove is that any countable representation of this class contains a class with infinite effective label space.

**Proposition 1.5.** *There exists a hypothesis class $\mathcal{H}_1 \subseteq \mathcal{Y}^{\mathcal{X}}$ that is universal multiclass online learnable under bandit feedback, but not non-uniform multiclass PAC learnable under full supervision. Also, there exists a hypothesis class $\mathcal{H}_2 \subseteq \mathcal{Y}^{\mathcal{X}}$ that is non-uniform multiclass online learnable under bandit feedback but not uniform multiclass PAC learnable under full supervision.*

The above proposition demonstrates a separation between uniform, non-uniform, and universal learning frameworks, particularly under bandit feedback and full supervision.

## 2 Related Work

**PAC Learning.** The Probably Approximately Correct (PAC) learning framework, introduced by Valiant [1984], has been a cornerstone in the field of statistical learning theory. Blumer et al. [1989], Vapnik and Chervonenkis [2015], Valiant [1984], Vapnik [2006] characterizes learnable classes within the binary PAC learning framework in the realizable setting via a combinatorial parameter called the VC dimension. This result was later extended to the agnostic setting by Haussler [1992]. Since then, PAC learning has been extensively studied in various learning theoretic settings.

**Online Learning.** Online learning has been a subject of study for over half a century. The seminal work of Littlestone [1988] marked the beginning of its formal exploration within the computer science community. Since then, online learning has been studied across diverse settings, including learning under bandit feedback Daniely et al. [2011], Daniely and Helbertal [2013], Long [2017], Geneson [2021], Raman et al. [2023], Hanneke and Yang [2023]. Additionally, it is closely linked to a wide set of fundamental problems, such as differential privacy, as explored in Alon et al. [2019], Bun et al. [2020], Alon et al. [2022]. Given its foundational nature, it is not surprising that online learning has also found many practical applications.

**Bandit Feedback.** The bandit setting plays a pivotal role in statistical decision-making. Its conceptual foundation was first established by Thompson [1933]. Later, the field was popularized by the mathematician and statistician Herbert Robbins, whose seminal work Robbins [1952] introduced the multi-armed bandit problem. In recent years, bandit scenarios have attracted substantial attention, driven by foundational contributions such as Auer et al. [2002a,b]. For a comprehensive review of the literature, the reader may refer to the recent work of Foster et al. [2021].

**Multiclass Learning.** A substantial body of theoretical research has explored multiclass classification in various frameworks, with notable contributions from Natarajan and Tadepalli [1988], Natarajan [1989], Ben-David et al. [1992], Haussler and Long [1995], Rubinstein et al. [2006], Daniely et al. [2011, 2012], Daniely and Shalev-Shwartz [2014], Brukhim et al. [2021]. However, a combinatorial characterization of multiclass classification within Valiant's PAC learning framework, particularly when the number of labels is unbounded, remained unresolved until recently, even in the realizable setting. This gap was addressed in the seminal work of Brukhim et al. [2022], which also extended to the agnostic setting as well David et al. [2016]. The study of multiclass classification with unbounded label spaces is motivated by several factors. First, in multiclass settings, it is preferable for guarantees to remain independent of the number of labels, even when finite. Second, mathematical frameworks involving infinity mostly provide clearer insights.

## 3 Notations, Definitions, and Preliminaries

In this section, we provide the model setting and formal definition of the problem.

**PAC Learning.** In this paper, we investigate the PAC learning with bandit feedback, which is defined in the work of Daniely et al. [2011], in a non-uniform setting. Formally, $\mathcal{X}$ is a non-empty *instance space* and $\mathcal{Y}$ is a non-empty *label space*. $\mathcal{H} \subseteq \mathcal{Y}^{\mathcal{X}}$ is the hypothesis class, which is a non-empty set of functions from $\mathcal{X}$ to $\mathcal{Y}$. In this paper, we focus on learning under the 0-1 loss. In other words, the loss function is $\mathbb{I}[y \neq y']$ defined in $\mathcal{Y} \times \mathcal{Y}$ and $\mathbb{I}[\cdot]$ is the indicator function. For a distribution $\mathcal{D}$ over $\mathcal{X} \times \mathcal{Y}$, we define the error of a concept $h$ with respect to the distribution $\mathcal{D}$ as $\mathrm{err}_{\mathcal{D}}(h) = \mathbf{Pr}_{(x,y) \sim \mathcal{D}}[h(x) \neq y]$. We also denote $\mathrm{err}_{\mathcal{D}}(\mathcal{H}) = \inf_{h \in \mathcal{H}} M_{\mathcal{D}}(h)$. A learning algorithm $\mathbf{A}$ under this setting is a function $\mathbf{A} : \cup_{n=1}^{\infty}(\mathcal{X} \times \mathcal{Y})^n \to \mathcal{Y}^{\mathcal{X}}$. It takes multiple samples, i.i.d. sampled from distribution $\mathcal{D}$, $S_m$, then outputs a concept $\hat{h}$. For the PAC learning with bandit feedback version, we first generate multiple samples, i.i.d. sampled from distribution $\mathcal{D}$, then the learner sees the instance sequence without labels and makes a prediction for each instance. After that, whether this prediction is correct is sent to the learner as feedback. Then the learner can use that information to output the concept $\hat{h}$. To unify the definitions, we also use $S_m$ to stand for the samples and feedback in the bandit feedback setting.

If there is a concept $h \in \mathcal{H}$, such that $\mathrm{err}_{\mathcal{D}}(h) = 0$, we say the distribution $\mathcal{D}$ is realizable by $\mathcal{H}$. Then we can define the non-uniform PAC learnability as follows.

**Definition 3.1** (Non-Uniform PAC Learnability). If for every $h^* \in \mathcal{H}$, there is a number $m(h^*, \epsilon, \delta)$, such that for every $m > m(h^*, \epsilon, \delta)$, we have $\mathbf{Pr}_{S_m \sim \mathcal{D}^m}[\mathrm{err}_{\mathcal{D}}(\mathbf{A}(S_m)) > \epsilon] \leq \delta$. We say $\mathcal{H}$ is non-uniform $(\epsilon, \delta)$-PAC learnable.

Then similarly, for the agnostic case,

**Definition 3.2** (Agnostic Non-Uniform PAC Learnability). If for every $h^* \in \mathcal{H}$, there is a number $m(h^*, \epsilon, \delta)$, such that for every $m > m(h^*, \epsilon, \delta)$, we have $\mathbf{Pr}_{S_m \sim \mathcal{D}^m}[\mathrm{err}_{\mathcal{D}}(\mathbf{A}(S_m)) > \mathrm{err}_{\mathcal{D}}(\mathcal{H}) + \epsilon] \leq \delta$. We say $\mathcal{H}$ is agnostic non-uniform $(\epsilon, \delta)$-PAC learnable.

**Online Learning.** In this paper, we investigate multiclass online learning in a non-uniform setting. Here, the definition of $\mathcal{X}$, $\mathcal{Y}$ and $\mathcal{H}$ keeps the same as previous. We also focus on the 0-1 loss in this setting. $X = \{X_t\}_{t \in \mathbb{N}}$ is a sequence of instances and $Y = \{Y_t\}_{t \in \mathbb{N}}$ is a sequence of labels.

Online learning is a sequential game between the learner and the adversary. In round $t$, the adversary reveals an instance $X_t$ to the learner and the learner makes a prediction $\hat{Y}_t$ based on the history $(X_{\leq t-1}, \tilde{Y}_{\leq t-1}) = \{(X_i, \tilde{Y}_i)\}_{i \leq t-1}$, where $\tilde{Y}_i$ is the feedback given to the learner. After that, the adversary gives the learner feedback $\tilde{Y}_t$ of its prediction, which may be used to inform future predictions. In this paper, we focus on two types of feedback: *full supervision* and *bandit feedback*. In the full supervision setting, the learner receives the true label $Y_t$ of the instance $X_t$ and incurs a loss $\mathbb{I}\left[Y_t \neq \hat{Y}_t\right]$. In the bandit feedback setting, the learner only receives the loss $\mathbb{I}\left[Y_t \neq \hat{Y}_t\right]$ and does not observe the true label $Y_t$.

In the realizable setting, the adversary chooses a concept $h^* \in \mathcal{H}$, which is unknown to the learner, and generates the labels $Y_t = h^*(X_t)$. The learner aims to minimize the cumulative number of mistakes,

$$\mathbf{M_A}(h^*, X, T) := \sum_{t=1}^{T} \mathbb{I}\left[\hat{Y}_t \neq h^*(X_t)\right].$$

We use $\mathbf{M_A}(h^*, X)$ to stand for $\mathbf{M_A}(h^*, X, \infty)$.

Unlike the classic online learning problem, we aim to build a mistake bound for each concept instead of a uniform mistake bound for the whole concept class. Formally speaking,

**Definition 3.3** (Non-uniform Online Learnability.). We say a concept class $\mathcal{H}$ is non-uniform online learnable in the realizable setting, if there exists an online learning algorithm $\mathbf{A}$, such that

$$\exists m : \mathcal{H} \to \mathbb{N}, \forall h^* \in \mathcal{H}, \forall X, \mathbf{M_A}(h^*, X) \leq m(h^*).$$

In the agnostic setting, we release the restriction that the adversary chooses a target concept from the concept class. Instead, the adversary can choose any sequence of labels $Y \in \mathcal{Y}^{\infty}$. In this case, we

use the notion of *regret* to measure the performance of the learner. The regret of the learner compared to a concept $h^*$ is defined as the difference between the total loss of the learner and the total loss of the function $h^*$. Formally speaking,

$$\text{regret}_{\mathbf{A}}(h^*, X, Y, T) := \mathbb{E}\left[\sum_{t=1}^{T} \mathbb{I}\left[\hat{Y}_t \neq Y_t\right] - \sum_{t=1}^{T} \mathbb{I}[h^*(X_t) \neq Y_t]\right].$$

**Remark.** Here the expectation is taken over the randomness of the learner. We do not need to consider the randomized learner for the realizable cases.

Then, we can define online learnability in the agnostic setting as follows.

**Definition 3.4** (Agnostic Non-Uniform Online Learnability.)**.** We say a concept class $\mathcal{H}$ is agnostic non-uniform online learnable, if there exists an online learning algorithm $\mathbf{A}$, such that

$$\exists m : \mathcal{H} \times \mathbb{N} \to \mathbb{N}, m(h^*, T) = o(T),$$
$$\forall h^* \in \mathcal{H}, \forall X, \text{ and } \forall Y, \text{regret}_{\mathbf{A}}(h^*, X, Y, T) \leq m(h^*, T).$$

We then introduce several useful combinatorial dimensions to describe our characterization.

## 3.1 Combinatorial Complexity Parameters

In this paper, we define $\mathcal{H}|_S = \{(y_1, \ldots, y_n) : \exists h \in \mathcal{H}, \forall i \leq n, y_i = h(x_i)\}$, where $S = (x_1, \ldots, x_n)$.

**Definition 3.5** (Pseudo-cube[Brukhim et al., 2022])**.** A class $\mathcal{H} \subseteq \mathcal{Y}^d$ is called a *pseudo-cube* of dimension $d$ if it is non-empty, finite and for every $h \in \mathcal{H}$ and $i \in [d]$, there is an $i$-neighbor $g \in \mathcal{H}$ of $h$ (i.e., $g(i) \neq h(i)$ and $g(j) = h(j)$ for all $j \neq i$).

**Definition 3.6** (DS dimension [Daniely and Shalev-Shwartz, 2014])**.** We say that $S \in \mathcal{X}^n$ is *DS-shattered* by $\mathcal{H} \subseteq \mathcal{Y}^{\mathcal{X}}$ if $\mathcal{H}|_S$ contains an $n$-dimensional pseudo-cube. The DS dimension $d_{DS}(\mathcal{H})$ is the maximum size of a DS-shattered sequence.

**Definition 3.7** (Natarajan dimension [Natarajan, 1989])**.** We say that $S \in \mathcal{X}^n$ is *N-shattered* by $\mathcal{H} \subseteq \mathcal{Y}^{\mathcal{X}}$ if there exist $f, g : [n] \to \mathcal{Y}$ such that for every $i \in [n]$ we have $f(i) \neq g(i)$, and

$$\mathcal{H}|_S \supseteq \{f(1), g(1)\} \times \{f(2), g(2)\} \times \ldots \times \{f(n), g(n)\}.$$

The Natarajan dimension $d_N(\mathcal{H})$ is the maximum size of an N-shattered sequence.

**Definition 3.8** (Multiclass Littlestone Tree and Littlestone Dimension [Daniely et al., 2011])**.** *A multiclass Littlestone tree* for a concept class $\mathcal{H}$ is a perfect binary tree with depth $d \leq \infty$. The internal nodes of that tree are labeled with elements of $\mathcal{X}$ and the edges connecting a node and its two children are labeled by two different labels from $\mathcal{Y}$, so that each finite path emanating from the root is consistent with a concept $h \in \mathcal{H}$. (Meaning that for each non-leaf node on the path, the concept $h$ labels the corresponding element of $\mathcal{X}$ with the label of the edge the path follows from that node.)

We say the Littlestone dimension of a concept class $\mathcal{H}$, $\text{Ldim}(\mathcal{H}) = d$, if there exists a multiclass Littlestone tree of depth $d$ shattered by $\mathcal{H}$, but there is no multiclass Littlestone tree of depth $d + 1$ shattered by $\mathcal{H}$.

**Definition 3.9** (Bandit Littlestone Tree and Bandit Littlestone Dimension [Daniely et al., 2011])**.** *A bandit Littlestone tree* (BL-tree for brevity) for a concept class $\mathcal{H}$ is a perfect tree $T$, with depth $d < \infty$. The internal nodes of that tree are labeled with elements of $\mathcal{X}$ and the edges connecting a node and its children are labeled with elements of $\mathcal{Y}$, without repetition. By saying a tree "perfect", we mean that for every element $y \in \mathcal{Y}$, there exists an edge labeled $y$. We say that a BL tree $T$ is shattered by a concept class $\mathcal{H}$, if for every path $P$ from the root to any leaf of $T$, there exists a concept $h \in \mathcal{H}$ such that for every internal node $v$ in $P$, $h(x_v)$ is not equal to the label of the edge connecting $v$ and its child.

The bandit Littlestone dimension of a concept class $\mathcal{H}$, $\text{BLdim}(\mathcal{H}) = d$, if there exists a bandit Littlestone tree of depth $d$ for $\mathcal{H}$, but there is no bandit Littlestone tree of depth $d + 1$ for $\mathcal{H}$.

If a concept class $\mathcal{H}$ has a finite (bandit) Littlestone dimension, we call it a (bandit) Littlestone class.

# 4 PAC Learning

In this section, we investigate the non-uniform PAC multiclass learning with bandit feedback. Basically, we combine the tools from the work of Benedek and Itai [1988] and the work of Daniely et al. [2011] to get a non-uniform PAC learner for the bandit feedback setting.

First, we show a property of the Natarajan dimension and the size of the effective label space. The proof appears in Appendix A.

**Lemma 4.1.** *If $\mathcal{H}_1$ has the size of effective label space $k_1$ and $d_N(\mathcal{H}_1) = d_1$, and $\mathcal{H}_2$ has the size of effective label space $k_2$ and $d_N(\mathcal{H}_2) = d_2$. Let $\mathcal{H} = \mathcal{H}_1 \cup \mathcal{H}_2$, the size of the effective label space of $\mathcal{H}$ is at most $k_1 + k_2$ and $d_N(\mathcal{H}) \leq (d_1 + d_2)^2 + 1$.*

To prove the main theorem in this section, we need the following result from Daniely et al. [2011].

**Theorem 4.2** ([Daniely et al., 2011]). *Let $\mathcal{H} \subseteq \mathcal{Y}^{\mathcal{X}}$ be a hypothesis class. $\mathcal{H}$ is uniformly PAC learnable with bandit feedback, if and only if $\mathcal{H}$ has finite Natarajan dimension and finite size of effective label space. More specifically, if the size of the effective label space of $\mathcal{H}$ is $k$. Then,*

$$m_b^r(\epsilon, \delta) = O\left( k \cdot \frac{d_N(\mathcal{H}) \log k \cdot \ln\left(\frac{1}{\epsilon}\right) + \ln(\frac{1}{\delta})}{\epsilon} \right) \ and \ m_b^a(\epsilon, \delta) = O\left( k \cdot \frac{d_N(\mathcal{H}) \log k + \ln(\frac{1}{\delta})}{\epsilon^2} \right).$$

*Here, $m_b^r(\epsilon, \delta)$ is the number of samples required for the realizable case, and $m_b^a(\epsilon, \delta)$ is the number of samples required for the agnostic case.*

*Proof of Theorem 1.3.* In this proof, we provide a way to transform a uniform PAC learning algorithm to a non-uniform PAC learning algorithm. As this method can be used to realizable algorithms and agnostic algorithms, we use the realizable case as an example in the proof.

First, we show that if $\mathcal{H}$ is non-uniform PAC learnable, $\mathcal{H}$ can be represented as a countable union of hypothesis classes with finite effective label space and Natarajan dimension. By definition, for every $h^* \in \mathcal{H}$, there is a number $m(h^*, \epsilon, \delta)$, such that if we have more than that number of samples, we can learn that concept. Therefore, by taking $\epsilon = \frac{1}{100}$ and $\delta = \frac{1}{100}$, we can set the following concept classes:

$$\mathcal{H}_i = \{h^* : m(h^*, \frac{1}{100}, \frac{1}{100}) \leq i\}$$

For every $i$, we know that there is a learning algorithm which only takes finite samples and can learn $h \in \mathcal{H}_i$ with $\epsilon = \frac{1}{100}$ and $\delta = \frac{1}{100}$. Due to Theorem 4.2, we know $\mathcal{H}_i$ has finite Natarajan dimension and finite size of effective label space. And also we know that for every $h \in \mathcal{H}$, there is an $i$, $h \in \mathcal{H}_i$, thus, $\mathcal{H} = \bigcup_i \mathcal{H}_i$.

Then we show how to non-uniformly learn a concept class $\mathcal{H}$ if it is a countable union of hypothesis classes with finite effective label space and Natarajan dimension. Assume $\mathcal{H} = \bigcup_i \mathcal{H}_i$. Due to Lemma 4.1, we assume the size of effective label space of $\bigcup_{i=1}^{j} \mathcal{H}_i$ is $k_j$ and $d_N(\bigcup_{i=1}^{j} \mathcal{H}_i) = d_j$. Then we describe the non-uniform learning algorithm. If we are given $m$ samples, compute the largest $j$, such that $m \geq k_j \cdot j(d_j \log k_j \cdot \ln j + \ln j)$, then we run the uniform learner on $\bigcup_{i=1}^{j} \mathcal{H}_i$ with $m$ samples, if there is a concept consistent with all samples, output it. Otherwise, output a prescribed concept $h_0$.

Then we show the algorithm works. For every concept $h^* \in \mathcal{H}$, we know there is a $j$ such that $h^* \in \mathcal{H}_j$. According to the algorithm above, we have $m(h^*, \epsilon, \delta) = O(k_{j'} \cdot j'(d_{j'} \log k_{j'} \cdot \ln j' + \ln j'))$, where $j' = \max j, \frac{1}{\epsilon}, \frac{1}{\delta}$. $\qquad\square$

# 5 Adversarial Online Learning

In this section, we investigate the non-uniform online multiclass learning in the realizable setting. By using the generic non-uniform learning algorithm and the bandit standard optimal algorithm (BSOA) from the work of Daniely et al. [2011], we prove that a concept class $\mathcal{H}$ is non-uniform online learnable if and only if it can be represented as a countable union of bandit Littlestone classes.

*Proof of Theorem 1.1.* We first prove that if $\mathcal{H}$ can be represented as a countable union of bandit Littlestone classes, then $\mathcal{H}$ is non-uniform online learnable with bandit feedback by using the

---
**Algorithm 1** Generic Non-uniform Online Learning Algorithm
---
1: **Input:** A concept class $\mathcal{H} = \bigcup_{n \in \mathbb{N}^+} \mathcal{H}_n$ with $d_n = \text{BLdim}(\mathcal{H}_n) < \infty, \forall n \in \mathbb{N}^+$.
2: Initialize a BSOA $\mathbf{A}_n$ for each bandit Littlestone class $\mathcal{H}_n$, and number of mistakes for each
   algorithm $e_n = 0$.
3: **for** $t = 1, 2, \ldots$ **do**
4:      Receive instance $X_t$.
5:      Compute $J_t = \arg\min_n\{e_n + n\}$.
6:      Predict $\hat{Y}_t$ as $\mathbf{A}_{J_t}$.
7:      Receive feedback $\mathbb{I}\left[Y_t \neq \hat{Y}_t\right]$.
8:      **if** $\mathbb{I}\left[Y_t \neq \hat{Y}_t\right] = 1$ **then**
9:         $e_{J_t} = e_{J_t} + 1$.
10:     **end if**
11: **end for**
---

following generic non-uniform online learning algorithm (Algorithm 1). Recall the result from the work of Daniely et al. [2011] that if a concept class $\mathcal{H}$ has $\text{BLdim}(\mathcal{H}) = d$, the BSOA has a mistake bound of $d$. Then we can prove that Algorithm 1 has a mistake bound of $m(h^*)$ is $O((k + d_k)^2)$, if $h^* \in \mathcal{H}_k$ and $\text{BLdim}(\mathcal{H}_k) = d_k$. Notice that for all $t$, $J_t + e_{J_t} \leq k + d_k$. Also, notice that the number of mistakes made by Algorithm1 is equal to the sum of the number of mistakes made by each $\mathbf{A}_n$. That is,

$$\mathbf{M}_A(h^*, X) = \sum_{i=1}^{\infty} e_i = \sum_{i=1}^{k+d_k} e_i + \sum_{i=k+d_k+1}^{\infty} e_i \leq (k + d_k)^2 + 0 = (k + d_k)^2.$$

We then prove that if $\mathcal{H}$ is non-uniform online learnable with bandit feedback, then $\mathcal{H}$ can be represented as a countable union of bandit Littlestone classes. Because $\mathcal{H}$ is non-uniform online learnable with bandit feedback, there exists an online learning algorithm $\mathbf{A}$ such that for all $h^* \in \mathcal{H}$, $\mathbf{M_A}(h^*, X) \leq m(h^*)$. Then we can define $\mathcal{H}_n = \{h \in \mathcal{H} : m(h) = n\}$. Because $\mathcal{H}$ is non-uniform online learnable with bandit feedback, we know that for every $h \in \mathcal{H}$, $m(h)$ is finite, and we can assume that the learning algorithm is $\mathbf{A}$. Thus, $\mathcal{H} = \bigcup_{n \in \mathbb{N}} \mathcal{H}_n$. We then prove that $\mathcal{H}_n$ is a bandit Littlestone class. In order to do this, we need the result from the work of Daniely et al. [2011] that any deterministic algorithm makes at least $\text{BLdim}(\mathcal{H})$ mistakes against the worst adversary for concept class $\mathcal{H}$. Notice that for any concept $h^* \in \mathcal{H}_n$, $\mathbf{A}$ can learn it with at most $n$ mistakes, thus, $\text{BLdim}(\mathcal{H}_n) \leq n$. Therefore, $\mathcal{H}_n$ is a bandit Littlestone class. $\qquad \square$

We provide the results for the agnostic setting, which is more technical in Appendix A.

## 6   Conclusion, Discussion, and Future Directions

In this paper, we study the fundamental problem of multiclass learning under bandit feedback when the number of labels can be unbounded within the non-uniform learning framework in both PAC and online models, in the realizable and the agnostic settings. We have shown a first theoretical result in this context, stating that PAC learnability is possible if and only if the hypothesis class can be represented as a countable union of classes with finite effective label space and finite Natarajan dimension, and online learnability is possible if and only if the hypothesis class can be represented as a countable union of classes with finite bandit Littlestone dimension. This includes elementary and natural hypothesis classes, such as the class of a countably infinite collection of constant functions over some domain, which is not learnable in the uniform learning framework. In addition, we provide a construction of a hypothesis class that is uniform online learnable under full supervision but not non-uniform PAC learnable under bandit feedback. This implies a key distinction between the non-uniform and universal learning frameworks.

At this point, we outline a suggestion for future research. We believe exploring alternative forms of feedback beyond bandit feedback in multiclass learning within the non-uniform learning framework is an avenue for future research. Do we have such a difference between uniform, non-uniform, and universal learning frameworks in any of them?

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

# A  Remaining Proofs

*Proof of Lemma 4.1.* First, it is obvious that the size of the effective label space of $\mathcal{H}$ is smaller than or equal to $k_1 + k_2$, because otherwise, there is a function $h \in \mathcal{H}$ and a instance $x$ such that $h(x) \neq h'(x)$ for every $h' \in \mathcal{H}_1$ and $h' \in \mathcal{H}_2$. Here is an obvious contradiction.

Then we prove the statement about Natarajan dimension by contradiction. Suppose $d_N(\mathcal{H}) > (d_1 + d_2)^2 + 1$. Thus, we have a sequence $S = (x_1, x_2, \ldots, x_{(d_1+d_2)^2+1}, x_{(d_1+d_2)^2+2})$ that is N-shattered by $\mathcal{H}$. Therefore, there are at least $2^{(d_1+d_2)^2+2}$ different label sequences in the intersection of $\mathcal{H}|_S$ and $\{y_1^0, y_1^1\} \times \{y_2^0, y_2^1\} \times \ldots \times \{y_{(d_1+d_2)^2+2}^0, y_{(d_1+d_2)^2+2}^1\}$, where $y_i^0 \neq y_i^1$ for all $i$. However, there are at most $\binom{(d_1+d_2)^2+2}{\leq d_1}$ different label sequences in the intersection of $\mathcal{H}_1|_S$ and $\{y_1^0, y_1^1\} \times \{y_2^0, y_2^1\} \times \ldots \times \{y_{(d_1+d_2)^2+2}^0, y_{(d_1+d_2)^2+2}^1\}$, and there are at most $\binom{(d_1+d_2)^2+2}{\leq d_2}$ different label sequences in the intersection of $\mathcal{H}_2|_S$ and $\{y_1^0, y_1^1\} \times \{y_2^0, y_2^1\} \times \ldots \times \{y_{(d_1+d_2)^2+2}^0, y_{(d_1+d_2)^2+2}^1\}$. Notice that $\binom{(d_1+d_2)^2+2}{\leq d_1} + \binom{(d_1+d_2)^2+2}{\leq d_2} \leq ((d_1+d_2)^2+2)^{d_1} + ((d_1+d_2)^2+2)^{d_2} < 2^{(d_1+d_2)^2+2}$. Therefore, it is a contradiction. $\qquad\square$

## A.1  Agnostic Setting

In this section, we discuss the result for the agnostic setting. We use the uniform online multiclass learning algorithm with bandit feedback from the work of Raman et al. [2023] to design a non-uniform online learning algorithm with bandit feedback. We then prove that a concept class $\mathcal{H}$ is non-uniform online learnable in the agnostic setting if and only if it can be represented as a countable union of bandit Littlestone classes. Formally,

**Theorem A.1.** *The following two statements are equivalent:*

- *$\mathcal{H}$ is non-uniform online learnable with bandit feedback in the agnostic setting.*

- *$\mathcal{H}$ can be represented as a countable union of bandit Littlestone classes.*

In order to prove this theorem, we need the following theorem from the work of Raman et al. [2023]:

**Theorem A.2** ([Raman et al., 2023], Theorem 2). *For any $\mathcal{H} \subseteq \mathcal{Y}^{\mathcal{X}}$, there exists an agnostic online learner whose expected regret, under bandit feedback, is at most*

$$8\sqrt{Ldim(\mathcal{H})BLdim(\mathcal{H})T \log T}.$$

To simplify this upper bound, we want to show $\mathrm{Ldim}(\mathcal{H}) \leq \mathrm{BLdim}(\mathcal{H})$ for all $\mathcal{H}$. This is true because we can transfer a Littlestone tree to a bandit Littlestone tree using the following method. For each node $v$ in the Littlestone tree, if its two children are $v_1$ and $v_2$ and the label of the two edges are $y_1$ and $y_2$, we can create a node $v$ in the bandit Littlestone tree and then add nodes $v_y$ for every $y \in \mathcal{Y}$ and label edge $(v, v_y)$ with $y$. Then we label $v_{y_1} = v_1$ and $v_{y_2} = v_2$, for other $y \in \mathcal{Y}$, we label $v_y = v_1$. Then we can prove that this bandit Littlestone tree can be shattered by $\mathcal{H}$. Thus, we have $\mathrm{Ldim}(\mathcal{H}) \leq \mathrm{BLdim}(\mathcal{H})$ for all $\mathcal{H}$. Therefore, the regret bound for the agnostic online learner with bandit feedback is upper bounded by $8\mathrm{BLdim}(\mathcal{H})\sqrt{T \log T}$.

Then we need the following lemma:

**Lemma A.3.** *If $BLdim(\mathcal{H}_1) = d_1$, $BLdim(\mathcal{H}_2) = d_2$, $BLdim(\mathcal{H}_1 \cup \mathcal{H}_2) \leq d_1 + d_2 + 2$.*

*Proof.* Suppose for sake of contradiction that $\mathrm{BLdim}(\mathcal{H}_1 \cup \mathcal{H}_2) \geq d_1 + d_2 + 3$. Then there exists a bandit Littlestone tree $T$ of depth $d_1 + d_2 + 3$ that can be shattered by $\mathcal{H}_1 \cup \mathcal{H}_2$. Then consider all subtrees at depth $d_1 + 1$, all of those subtrees have depth $d_2 + 2$. Then because the bandit Littlestone dimension of $\mathcal{H}_2$ is $d_2$, the version space at the root of the subtree must contain some $h \in \mathcal{H}_1$. Therefore, the subtree from the root to every node at the depth $d_1 + 1$ can be shattered by $\mathcal{H}_1$, which contradicts the fact that $\mathrm{BLdim}(\mathcal{H}_1) = d_1$. That finishes the proof. $\qquad\square$

Then we can use the above lemmas and theorems to prove Theorem A.1.

*Proof of Theorem A.1.* We first show that if $\mathcal{H}$ can be represented as a countable union of bandit Littlestone classes, then $\mathcal{H}$ is non-uniform online learnable with bandit feedback in the agnostic

setting. First, as $\mathcal{H}$ can be represented as a countable union of bandit Littlestone classes, we can represent it as the following way: $\mathcal{H} = \bigcup_{i=1}^{\infty} \mathcal{H}_i$, such that $\mathrm{BLdim}(\bigcup_{i=1}^{k} \mathcal{H}_i) \leq k$. Then we can use the doubling trick to handle the changing time horizon. Set $T_k = \sum_{j=1}^{k} 2^{j-1}$, then run the non-uniform online learning algorithm with bandit feedback in the agnostic setting with reference concept class $\bigcup_{i=1}^{k} \mathcal{H}_i$ in the time slot $[T_k, T_{k+1}]$.

Then we analyze this algorithm $\mathbf{A}$, notice that for every $h^*$, there is a $k$, such that $h^* \in \mathcal{H}_k$. In the meanwhile, by using Theorem A.2, we can get a regret bound for each time slot. Thus, we can compute the regret of the algorithm as follows:

$$
\begin{aligned}
&\mathrm{regret}_{\mathbf{A}}(h^*, X, Y, T) \\
&= \mathbb{E}\left[ \sum_{t=1}^{T} \mathbb{I}\left[\hat{Y}_t \neq Y_t\right] - \sum_{t=1}^{T} \mathbb{I}[h^*(X_t) \neq Y_t] \right] \\
&= \sum_{j=1}^{\log T} \mathbb{E}\left[ \sum_{t=T_j+1}^{T_{j+1}} \mathbb{I}\left[\hat{Y}_t \neq Y_t\right] - \sum_{t=T_j+1}^{T_{j+1}} \mathbb{I}[h^*(X_t) \neq Y_t] \right] \\
&\leq T_k + \sum_{j=k}^{\lfloor \log T \rfloor} \mathbb{E}\left[ \sum_{t=T_j+1}^{T_{j+1}} \mathbb{I}\left[\hat{Y}_t \neq Y_t\right] - \mathbb{I}[h^*(X_t) \neq Y_t] \right] \\
&\quad + \mathbb{E}\left[ \sum_{t=T_{\lfloor \log T \rfloor}+1}^{T} \mathbb{I}\left[\hat{Y}_t \neq Y_t\right] - \mathbb{I}[h^*(X_t) \neq Y_t] \right] \\
&\leq T_k + \sum_{j=k}^{\lfloor \log T \rfloor} 8\mathrm{BLdim}\left( \bigcup_{i=1}^{j} \mathcal{H}_i \right) \sqrt{j 2^j} \\
&\quad + 8\mathrm{BLdim}\left( \bigcup_{i=1}^{\lfloor \log T \rfloor+1} \mathcal{H}_i \right) \sqrt{(T - 2^{\lfloor \log T \rfloor})\log(T - 2^{\lfloor \log T \rfloor})} \\
&= T_k + \sum_{j=k}^{\lfloor \log T \rfloor} 8j\sqrt{j 2^j} \\
&\quad + 8(\lfloor \log T \rfloor + 1)\sqrt{(T - 2^{\lfloor \log T \rfloor})\log(T - 2^{\lfloor \log T \rfloor})} \\
&\leq T_k + 8(\lfloor \log T \rfloor)^{\frac{5}{2}}\sqrt{2^{\lfloor \log T \rfloor+1}} + 8(\lfloor \log T \rfloor + 1)^{\frac{3}{2}}\sqrt{T} \\
&\leq T_k + 8(\lfloor \log T \rfloor + 1)^{\frac{5}{2}}\sqrt{T}
\end{aligned}
$$

Notice that $T_k$ is a constant related to $h^*$, thus, we have $m(h^*, T) = O(\sqrt{T \log^5 T})$, which is $o(T)$. Therefore, this algorithm is a non-uniform online learning algorithm with bandit feedback in the agnostic setting.

Then we need to prove that if $\mathcal{H}$ cannot be represented as a countable union of bandit Littlestone classes, no non-uniform online learning algorithm with bandit feedback can learn $\mathcal{H}$ in the agnostic setting. It is equivalent to prove that if $\mathcal{H}$ cannot be represented as a countable union of bandit Littlestone classes, no non-uniform online learning algorithm with bandit feedback can learn $\mathcal{H}$ with a $o(T)$ mistake bound, that is, for all $h^* \in \mathcal{H}$, $m(h^*, T) = o(T)$. We prove the contrapositive of this statement. In other words, if there is a non-uniform online learner $\mathbf{A}$ for a concept class $\mathcal{H}$ with mistake bound $o(T)$, $\mathcal{H}$ can be represented as a countable union of bandit Littlestone classes. By definition, we define

$$
\mathcal{H}_i = \{h \in \mathcal{H} : \forall X, \forall T \geq i, \mathbb{E}[\mathbf{M_A}(h, X, T)] \leq \frac{T}{3}\}.
$$

Notice that $i \in \mathbb{N}$ and $\mathcal{H} = \bigcup_{i=1}^{\infty} \mathcal{H}_i$. Define $C_i = \sup_x |\{h(x) : h \in \mathcal{H}_i\}|$. Then refer to the Lemma 18 in the work of Raman et al. [2023] and notice that we have $\mathbb{E}[\mathbf{M}_{A\mathbf{A}}(h, X, T)] \leq \frac{T}{3}$ for all

$T \geq i$ and all $X$. Thus, by taking $T = i$, we have the following two inequalities:

$$\frac{i}{3} \geq \frac{C_i - 1}{2}$$

$$\frac{i}{3} \geq \frac{\mathrm{BLdim}(\mathcal{H}_i)}{4 C_i \log C_i}$$

Therefore, we have $\mathrm{BLdim}(\mathcal{H}_i) \leq \frac{4i(2i+1)}{9} \log(\frac{2i+1}{3})$, which is finite. So, $\mathcal{H}_i$ is a bandit Littlestone class for all $i$, which implies that $\mathcal{H}$ can be represented as a countable union of bandit Littlestone classes. That finishes the proof. $\qquad\square$

## B  Examples

In this section, we provide three examples of hypothesis classes revealing separations between related learnability notions. The main innovation is constructing the first example. The second example is based on the work of Bousquet et al. [2021]. Also, the third example uses ideas from the second example.

**Proposition B.1** (Non-Uniform Multiclass Online Learnable **but not** Non-Uniform Multiclass PAC Learnable under Bandit Feedback). *There exists a hypothesis class $\mathcal{H} \subseteq \mathcal{Y}^{\mathcal{X}}$ such that can be represented as a countable union of classes with finite Littlestone dimension, but cannot be represented as a countable union of classes with finite effective label space.*

*Proof.* Let $\mathcal{X}$ be the set of all non-empty countable sequences of distinct real numbers. Also, let $\mathcal{Y} = \mathbb{N} \cup \{\star\}$. In addition, let $\mathcal{H} = \{h_y \mid h_y : \mathcal{X} \to \mathcal{Y}, \ y \in \mathbb{R}, \ \forall_{S \in \mathcal{X}} \ h_y(S) = \star$ if $y$ isn't in the sequence $S$; otherwise $h_y(S) = i$, where $S_i = y\}$. First, we claim that $\mathcal{H}$ cannot be represented as a countable union of classes with finite effective label space. Suppose by contradiction that $\mathcal{H} = \cup_{i=1}^{\infty} \mathcal{H}_i$ such that for every $i \in \mathbb{N}$, we have: the effective label space of $\mathcal{H}_i$ is finite. Then, there exists $i^\star \in \mathbb{N}$ such that $|\mathcal{H}_{i^\star}| = \infty$. This is because $\mathcal{H}$ is uncountable as $\mathbb{R}$ is uncountable. Now, we claim that the effective label space of $\mathcal{H}_{i^\star}$ is infinite. To see this, let $\mathcal{H}' = \{h_{z_1}, h_{z_2}, \dots\} \subseteq \mathcal{H}_{i^\star}$ such that $\mathcal{H}'$ is countable. Based on that, let $S' = (z_1, z_2, \dots)$. Then, observe that the set $\{y \mid y \in \mathbb{R}, \ \exists_{h \in \mathcal{H}'} h(S') = y\}$ is infinite. As a result, we should have: the effective label space of $\mathcal{H}_{i^\star}$ is infinite. This is a contradiction. Second, we claim that $\mathrm{Ldim}(\mathcal{H}) = 1$. To show this, we prove that the following online learning algorithm makes at most one mistake against any realizable adversary. The algorithm predicts $\star$ until the first mistake, after which there is only one concept consistent with the true label. This is because if the mistake was made on $(S, i)$, then the target function is $h_z$, where $z = S_i$. This finishes the proof. $\qquad\square$

**Proposition B.2** (Universal Multiclass Online Learnable under Bandit Feedback **but not** Non-Uniform Multiclass PAC Learnable). *There exists a hypothesis class $\mathcal{H} \subseteq \mathcal{Y}^{\mathcal{X}}$ such that it does not have an infinite Littlestone tree, but cannot be represented as a countable union of classes with finite VC dimension.*

*Proof.* Let $\mathcal{X} = \{S \mid S \subset \mathbb{R}, |S| < \infty\}$. Also, let $\mathcal{Y} = \{0, 1\}$. In addition, let $\mathcal{H} = \{h_y \mid h_y : \mathcal{X} \to \mathcal{Y}, y \in \mathbb{R}, \forall_{S \in \mathcal{X}} \ h_y(S) = \mathbb{I}[y \in S]\}$. First, we claim that $\mathcal{H}$ cannot be represented as a countable union of classes with finite VC dimension. Suppose by contradiction that $\mathcal{H} = \cup_{i=1}^{\infty} \mathcal{H}_i$ such that for every $i \in \mathbb{N}$, we have: $\mathrm{VC}(\mathcal{H}') < \infty$. Then, there exists $i^\star \in \mathbb{N}$ such that $|\mathcal{H}_{i^\star}| = \infty$. This is because $\mathcal{H}$ is uncountable as $\mathbb{R}$ is uncountable. Now, we claim that $\mathrm{VC}(\mathcal{H}_{i^\star}) = \infty$. This is because the dual class of $\mathcal{H}_{i^\star}$ is the class of all finite subsets of an uncountable set. In particular, note that for every hypothesis class $\mathcal{H}' \subseteq \{0, 1\}^{\mathcal{X}'}$, we have: $\mathrm{VC}(\mathcal{H}') < \infty$ if and only if $\mathrm{VC}^\star(\mathcal{H}') < \infty$. This is a contradiction. Second, we claim that $\mathcal{H}$ does not have an infinite Littlestone tree. This is because once we fix a root $S \in \mathcal{X}$ of a Littlestone tree, the class $\{h \mid h \in \mathcal{H}, h(S) = 1\}$ is finite, so the corresponding subtree must also be finite. This finishes the proof. $\qquad\square$

**Proposition B.3** (Non-Uniform Multiclass Online Learnable under Bandit Feedback **but not** Uniform Multiclass PAC Learnable). *There exists a hypothesis class $\mathcal{H} \subseteq \mathcal{Y}^{\mathcal{X}}$ such that it can be represented as a countable union of classes with finite bandit Littlestone dimension, but $\mathrm{VC}(\mathcal{H}) = \infty$.*

*Proof.* Let $\mathcal{X} = \{S \mid S \subset \mathbb{N}, |S| < \infty\}$. Also, let $\mathcal{Y} = \{0, 1\}$. In addition, let $\mathcal{H} = \{h_y \mid h_y : \mathcal{X} \to \mathcal{Y}, y \in \mathbb{N}, \forall_{S \in \mathcal{X}} \ h_y(S) = \mathbb{I}[y \in S]\}$. First, we claim that $\mathrm{VC}(\mathcal{H}) = \infty$. This is because the dual class of $\mathcal{H}$ is the class of all finite subsets of $\mathbb{N}$. In particular, note that for every hypothesis class $\mathcal{H}' \subseteq \{0, 1\}^{\mathcal{X}'}$, we have: $\mathrm{VC}(\mathcal{H}') < \infty$ if and only if $\mathrm{VC}^\star(\mathcal{H}') < \infty$. Second, we claim that $\mathcal{H}$ can be represented as a countable union of classes with finite bandit Littlestone dimension. This is because $\mathcal{H}$ is countable. In particular, note that each single hypothesis has both Littlestone dimension and bandit Littlestone dimension equal to zero. This finishes the proof. $\square$

We note that, above, we use $\mathrm{VC}^\star(.)$ as the dual VC dimension. We refer the readers to the work of Assouad [1983] for the proof of the result that we used.

## C  Non-Uniform Multiclass PAC Learning

In this section, we investigate the non-uniform PAC multiclass learning. Basically, we combine the tools from the work of Benedek and Itai [1988] and the work of Brukhim et al. [2022] to get a non-uniform PAC learner.

First, we have the following property of the DS dimension.

**Lemma C.1.** *If $d_{DS}(\mathcal{H}_1) = d_1$, $d_{DS}(\mathcal{H}_2) = d_2$, $d_{DS}(\mathcal{H}_1 \cup \mathcal{H}_2) \leq d_1 + d_2 + 1$.*

*Proof.* We can prove this by contradiction. Suppose $d_{DS}(\mathcal{H}_1 \cup \mathcal{H}_2) \geq d_1 + d_2 + 2$. We have $S = \{x_1, \ldots, x_{d_1+d_2+2}\}$, such that $\mathcal{H}|_S$ contains a $d_1 + d_2 + 2$-dimension pseudo-cube. Without loss of generality, assume $S_1 = \{x_1, \ldots, x_{d_1}\}$ such that $\mathcal{H}_1|_{S_1}$ contains a $d_1$-dimension pseudo-cube. However, for any $i = d_1 + 1, \ldots, d_1 + d_2 + 2$, let $S_1' = S_1 \cup \{x_i\}$, then $\mathcal{H}_1|_{S_1'}$ does not contain a $d_1 + 1$ pseudo-cube. Therefore, for all $h \in \mathcal{H}_1$, they all agrees on the label of $S_2 = \{x_{d_1+1,\ldots,d_1+d_2+2}\}$. However, $\mathcal{H}|_{S_2}$ contains a $d_2 + 2$ dimension pseudo-cube. Therefore, $\mathcal{H}_2|_{S_2} = \mathcal{H}|_{S_2} \setminus \mathcal{H}_1|_{S_2}$, which must contains a $d_2 + 1$-dimension pseudo cube. Here is a contradiction, and that finishes the proof. $\square$

In order to prove the main theorem in this section, Theorem 1.3. We need the following results about the uniform PAC learnability with bandit feedback.

**Theorem C.2** ([Brukhim et al., 2022]). *Let $\mathcal{H} \subseteq \mathcal{Y}^{\mathcal{X}}$ be a hypothesis class. $\mathcal{H}$ is uniformly PAC learnable, if and only if $\mathcal{H}$ has finite DS dimension.*

Then we have the main theorem here.

**Theorem C.3.** *The following three statements are equivalent:*

- *$\mathcal{H}$ can be represented as a countable union of hypothesis classes whose DS dimension is finite.*

- *$\mathcal{H}$ is non-uniform PAC learnable in the realizable setting.*

- *$\mathcal{H}$ is non-uniform PAC learnable in the agnostic setting.*

Recall the proof of Theorem 1.3, we only need Lemma C.1, and the rest of the proof is exactly the same.

## D  Non-Uniform Multiclass Adversarial Online Learning

In this section, we provide the results for non-uniform multiclass online learning with full supervision. The results here are new, but can easily be obtained by extending the methods in the work of Lu [2024].

### D.1  Realizable Setting

In the realizable setting, a concept class $\mathcal{H}$ is non-uniformly online learnable, if and only if $\mathcal{H}$ is a countable union of Littlestone classes. The proof extends the algorithm from the work of Lu [2024] to a multiclass setting. Formally,

**Theorem D.1.** *The following two statements are equivalent:*

- *$\mathcal{H}$ is non-uniform online learnable with full supervision.*

- *$\mathcal{H}$ can be represented as a countable union of Littlestone classes.*

*Proof.* We first prove that if $\mathcal{H}$ can be represented as a countable union of Littlestone classes, then $\mathcal{H}$ is non-uniform online learnable with full supervision by using the following non-uniform online learning algorithm (Algorithm 2). Recall that the SOA algorithm has a mistake bound of $d$ for a

---

**Algorithm 2** Non-uniform Online Learning Algorithm with Full Supervision

---

1: **Input:** A concept class $\mathcal{H} = \bigcup_{n \in \mathbb{N}^+} \mathcal{H}_n$ with $d_n = \text{BLdim}(\mathcal{H}_n) < \infty, \forall n \in \mathbb{N}^+$.
2: Initialize an SOA $\mathbf{A}_n$ for each bandit Littlestone class $\mathcal{H}_n$, and number of mistakes for each algorithm $e_n = 0$.
3: **for** $t = 1, 2, \ldots$ **do**
4:     Receive instance $X_t$.
5:     Compute $J_t = \arg\min_n\{e_n + n\}$.
6:     Predict $\hat{Y}_t$ as $\mathbf{A}_{J_t}$.
7:     Receive true label $Y_t$.
8:     **for** $n = 1, 2, \ldots$ **do**
9:         **if** The prediction of $\mathbf{A}_n$ is not equal to $Y_t$ **then**
10:             $e_n = e_n + 1$.
11:         **end if**
12:     **end for**
13: **end for**

---

Littlestone class $\mathcal{H}$ with $\text{Ldim}(\mathcal{H}) = d$. Then if the target concept $h^* \in \mathcal{H}_k$, where $\text{Ldim}(\mathcal{H}_k) = d_k$. Then notice that for all $t$, we have $J_t \leq d_k + k$. Thus, the algorithm will stick to the $\mathbf{A}_k$ after at most $(d_k + k)^2$ rounds and will make no more mistakes. So we have $m(h^*) \leq (d_k + k)^2$ and that finishes the proof of the first part.

Then we prove that if $\mathcal{H}$ is non-uniform online learnable with full supervision, then $\mathcal{H}$ can be represented as a countable union of Littlestone classes. Because $\mathcal{H}$ is non-uniform online learnable with full supervision, there exists an online learning algorithm $\mathbf{A}$ such that for all $h^* \in \mathcal{H}$, $\mathbf{M}_{\mathbf{A}}(h^*, X) \leq m(h^*)$. Then we can define

$$\mathcal{H}_n = \{h \in \mathcal{H} : m(h) = n\}.$$

Because $\mathcal{H}$ is non-uniform online learnable with full supervision, we know that for every $h \in \mathcal{H}$, $m(h)$ is finite, and we can assume that the learning algorithm is $\mathbf{A}$. Thus, $\mathcal{H} = \bigcup_{n \in \mathbb{N}^+} \mathcal{H}_n$. We then prove that $\mathcal{H}_n$ is a Littlestone class. In order to do this, we need the result from the work of Daniely et al. [2011] that any deterministic algorithm makes at least $\text{Ldim}(\mathcal{H})$ mistakes against the worst adversary for concept class $\mathcal{H}$. Notice that for any concept $h^* \in \mathcal{H}_n$, $\mathbf{A}$ can learn it with at most $n$ mistakes, thus, $\text{Ldim}(\mathcal{H}_n) \leq n$. Therefore, $\mathcal{H}_n$ is a Littlestone class. That finishes the proof. □

### D.2 Agnostic Setting

In the agnostic setting, a concept class $\mathcal{H}$ is non-uniformly online learnable, if and only if $\mathcal{H}$ is a countable union of Littlestone classes. We use the doubling trick on the uniform online learning algorithm to show this result. Formally,

**Theorem D.2.** *The following two statements are equivalent:*

- *$\mathcal{H}$ is non-uniform online learnable with full supervision in the agnostic setting.*

- *$\mathcal{H}$ can be represented as a countable union of Littlestone classes.*

In order to prove this theorem, we need the uniform online learning algorithm in the agnostic setting and the following theorem from the work of Hanneke et al. [2023].

**Theorem D.3** ([Hanneke et al., 2023], Theorem 4). *For any concept class $\mathcal{H}$ and $T > 2Ldim(\mathcal{H})$, we have an agnostic online learning algorithm, whose regret is at most*

$$O(\sqrt{Ldim(\mathcal{H})T \log T}).$$

We can use the doubling trick and the uniform online learning algorithm above to design the non-uniform online learning algorithm. To do this, we need to notice that the property of having a finite Littlestone dimension is closed under finite union. This is true due to the same reasoning as Lemma A.3. Then we can prove Theorem D.2.

*Proof.* We first show that if $\mathcal{H}$ can be represented as a countable union of Littlestone classes, $\mathcal{H}$ is non-uniform online learnable in the agnostic setting. Because $\mathcal{H}$ can be represented as a countable union of Littlestone classes, we can write $\mathcal{H} = \bigcup_{i=1}^{\infty} \mathcal{H}_i$, such that $\mathrm{Ldim}(\bigcup_{i=1}^{k} \mathcal{H}_i) \leq k$. Let $T_k = \sum_{j=1}^{k} 2^{j-1}$. Run the uniform agnostic online learning algorithm on $\bigcup_{i=1}^{k} \mathcal{H}_i$ in time slot $[T_k + 1, T_{k+1}]$, we restart the algorithm at time $T_k + 1$ for every $k$.

Notice that for every $h^*$, there is a $k$ such that $h^* \in \mathcal{H}_k$, and use Theorem D.3, we can analyze the regret of the algorithm above:

$$
\mathrm{regret}_{\mathbf{A}}(h^*, X, Y, T) = \mathbb{E}\left[\sum_{t=1}^{T} \mathbb{I}\left[\hat{Y}_t \neq Y_t\right] - \sum_{t=1}^{T} \mathbb{I}[h^*(X_t) \neq Y_t]\right]
$$

$$
= \sum_{j=1}^{\log T} \mathbb{E}\left[\sum_{t=T_j+1}^{T_{j+1}} \mathbb{I}\left[\hat{Y}_t \neq Y_t\right] - \sum_{t=T_j+1}^{T_{j+1}} \mathbb{I}[h^*(X_t) \neq Y_t]\right]
$$

$$
\leq T_k + \sum_{j=k}^{\lfloor \log T \rfloor} \mathbb{E}\left[\sum_{t=T_j+1}^{T_{j+1}} \mathbb{I}\left[\hat{Y}_t \neq Y_t\right] - \mathbb{I}[h^*(X_t) \neq Y_t]\right] + \mathbb{E}\left[\sum_{t=T_{\lfloor \log T \rfloor}+1}^{T} \mathbb{I}\left[\hat{Y}_t \neq Y_t\right] - \mathbb{I}[h^*(X_t) \neq Y_t]\right]
$$

$$
\leq T_k + \sum_{j=k}^{\lfloor \log T \rfloor} c\sqrt{\mathrm{Ldim}\left(\bigcup_{i=1}^{j} \mathcal{H}_i\right) j2^j} + c\sqrt{\mathrm{Ldim}\left(\bigcup_{i=1}^{\lfloor \log T \rfloor+1} \mathcal{H}_i\right)(T - 2^{\lfloor \log T \rfloor})\log(T - 2^{\lfloor \log T \rfloor})}
$$

$$
\leq T_k + \sum_{j=k}^{\lfloor \log T \rfloor} cj\sqrt{2^j} + c(\lfloor \log T \rfloor + 1)\sqrt{(T - 2^{\lfloor \log T \rfloor})}
$$

$$
\leq T_k + c(\lfloor \log T \rfloor + 1)^2 \sqrt{T}
$$

This inequality holds for some constant $c$. Thus, we notice that $m(h^*, T) = T_k + c(\log T)^2\sqrt{T}$, which is $o(T)$. Therefore, this is a non-uniform agnostic online learning algorithm.

To prove the necessity, it is equivalent to show that if there is a non-uniform online learner $\mathbf{A}$ can learn a concept class $\mathcal{H}$ with expected mistake bound $m(h, T) = o(T)$, $\mathcal{H}$ can be represented as a countable union of Littlestone classes.

By definition, we define

$$
\mathcal{H}_i = \{h \in \mathcal{H} : \forall X, \forall T \geq i, \mathbb{E}[\mathbf{M}_{\mathbf{A}}(h, X, T)] \leq \frac{T}{3}\}.
$$

Notice that $i \in \mathbb{N}$ and $\mathcal{H} = \bigcup_{i=1}^{\infty} \mathcal{H}_i$. Referring to the work of Ben-David et al. [2009], we know there exists a sequence $X$, such that $\mathbb{E}[\mathbf{M}_{\mathbf{A}}(h, X, T)] \geq \frac{\mathrm{Ldim}(\mathcal{H}_i)}{2}$ for all algorithm $\mathbf{A}$. Then we take $T = i$, we can get $\frac{i}{3} \geq \frac{\mathrm{Ldim}(\mathcal{H}_i)}{2}$. Thus, $\mathrm{Ldim}(\mathcal{H}_i) \leq \frac{2i}{3}$, which is finite. So, $\mathcal{H}_i$ is a Littlestone class for all $i$, which implies that $\mathcal{H}$ can be represented as a countable union of Littlestone classes. That finishes the proof. $\square$

