# OpenReview forum: "Non-Uniform Multiclass Learning with Bandit Feedback"
_NeurIPS.cc/2025/Conference — NeurIPS 2025 poster_

### Official Review · Reviewer_wMkv · 2025-06-27

**Clarity:** 4
**Significance:** 3
**Originality:** 2
**Rating:** 4
**Confidence:** 4

**Summary:**

Studies the non-uniform model for multiclass classification with unbounded number of labels under bandit feedback. Considers both PAC and online learning settings. Provides a characterization of learnability in realizable/agnostic versions of PAC and online settings. Further unlike universal learning, there is a separation between bandit feedback and full feedback in the non-uniform setting. The non-uniform setting is in between uniform and universal learning settings. In uniform setting, sample complexity/regret should only depend on the hypothesis class. In non-uniform, it can depend on hypothesis as well. In universal learning, it can depend on hypothesis and distribution/individual sequence as well.

**Questions:**

What is truly novel in your algorithms and proofs?
Why should the broader ML community be interested in these result as opposed to a more specialized theory-focused conference?

**Ethical Concerns:**

["NO or VERY MINOR ethics concerns only"]

**Final Justification:**

The authors clarified both the contribution and why it should appeal to the neurips community.

**Limitations:**

yes

**Quality:**

3

**Strengths And Weaknesses:**

Strengths:
Consider a very natural question that was left unaddressed by previous work. Last few years have seen a lot of work on unbounded label spaces and bandit feedback. Makes the paper timely.

Weakness:
Novelty is very limited. The characterization is exactly what one would expect. In standard online or PAC learning, we know that non-uniformly learnable classes are exactly those that are countable unions of uniformly learnable classes. This is exactly what the authors show here too: countable unions of classes that are learnable uniformly. While the paper fills a gap technically speaking, the answer and the methodology used to provide the answer does not provide sufficiently novel insights into learnability in general. As a result, the paper is not appropriate for a broad audience conference like NeurIPS. It might be more suitable for a theory focused conference. This is especially so since the focus on unbounded label spaces and the subtle differences between uniform and non-uniform learnability, while fully justified from a theoretical point of view, are of limited relevance to the broad ML community.

---

> ### Author Rebuttal · Authors · 2025-07-31
>
> We thank the reviewer for dedicating their time to assess our work. We are delighted that the reviewer found our paper timely, and moreover, mentioned that we considered a very natural question. Below, we address comments provided by the reviewer.
>
>
> > "What is truly novel in your algorithms and proofs?"
>
> > "Novelty is very limited. The characterization is exactly what one would expect. In standard online or PAC learning, we know that non-uniformly learnable classes are exactly those that are countable unions of uniformly learnable classes. This is exactly what the authors show here too: countable unions of classes that are learnable uniformly. While the paper fills a gap technically speaking, the answer and the methodology used to provide the answer does not provide sufficiently novel insights into learnability in general."
>
> We construct a hypothesis class that is non-uniformly learnable under full supervision in the adversarial online model (and thus also in the i.i.d. batch model), but not non-uniformly learnable under bandit feedback in the i.i.d. batch model (and thus also not in the adversarial online model). We strongly believe this construction is novel, as the hypothesis class differs significantly from natural or commonly considered classes. In summary, this serves as our main novel technical contribution that reveals a fundamental distinction between the non-uniform and universal learning frameworks.
>
>
> > "Why should the broader ML community be interested in these result as opposed to a more specialized theory-focused conference?"
> > "the paper is not appropriate for a broad audience conference like NeurIPS. It might be more suitable for a theory focused conference. This is especially so since the focus on unbounded label spaces and the subtle differences between uniform and non-uniform learnability, while fully justified from a theoretical point of view, are of limited relevance to the broad ML community."
>
> To the best of our knowledge, NeurIPS encompasses multiple sub-communities, each with specific interests that may not necessarily align with those of others. We believe that our results are particularly relevant and of interest to the ML theory sub-community within NeurIPS.
>
>
> Finally, once again, we thank the reviewer for dedicating their time to assess our work. We hope this rebuttal convinces you of the novelty of our contribution.

---

> > ### Comment · Reviewer_wMkv · 2025-08-01
> >
> > Thanks for that clarification. However, I still feel that a theory paper in NeurIPS should speak to an audience slightly larger than ML theory sub-community. Your comment about the novelty of the hypothesis class construction for separating full information and bandit setting is well received. But does it have any broader relevance beyond someone interested in both a) non-uniform setting and b) unbounded number of labels? My main concern is that the number of people interested in both a) and b) is quite small. I still see value in your results because they answer a natural question. But they question is only natural for someone interested in both a) and b). I maintain my assessment that the paper is weak for a broad audience conference like NeurIPS.

---

> ### Author Response · Authors · 2025-08-05
>
> We thank the reviewer for dedicating their time to reassess our work. Below, we address the new comment provided by the reviewer.
>
> We respectfully note that both topics (a) and (b) have been the focus of many papers in the recent history of NeurIPS. Indeed, they have also been the subject of considerable attention in the learning theory community, especially in recent years. For instance, Lu studied non-uniform online binary classification, presenting it as a spotlight talk at NeurIPS 2024 [1]. In close relation to the non-uniform learning framework, the following works on the universal learning framework [2, 3, 4, 5, 6] appeared at NeurIPS 2022 (oral), 2022, 2024, 2024, and 2024, respectively. In addition, there are several papers on unbounded label space, including [7, 8, 9, 10], which appeared at NeurIPS 2023, 2024 (spotlight), 2024, and 2024, respectively. Finally, we note that the notion of non-uniform learning has also recently appeared in the context of language generation theory. For example, see [11].
>
> We hope this rebuttal convinces you further of the relevance of our contributions to the NeurIPS community.
>
> Once again, thank you for dedicating your time to reassessing our work.
>
>
> 1. Lu, Z. When Is Inductive Inference Possible? In Advances in Neural Information Processing Systems, Vol. 37, NeurIPS 2024.
> 2. Hanneke, S., Karbasi, A., Moran, S., & Velegkas, G. Universal Rates for Interactive Learning. In Advances in Neural Information Processing Systems, Vol. 35, NeurIPS 2022.
> 3. Kalavasis, A., Velegkas, G., & Karbasi, A. Multiclass Learnability Beyond the PAC Framework: Universal Rates and Partial Concept Classes. In Advances in Neural Information Processing Systems, Vol. 35, NeurIPS 2022.
> 4. Hanneke, S., Karbasi, A., Moran, S., & Velegkas, G. Universal Rates for Active Learning. In Advances in Neural Information Processing Systems, Vol. 37, NeurIPS 2024.
> 5. Hanneke, S., & Xu, M. Universal Rates of Empirical Risk Minimization. In Advances in Neural Information Processing Systems, Vol. 37, NeurIPS 2024.
> 6. Hanneke, S., & Wang, H. A Theory of Optimistically Universal Online Learnability for General Concept Classes. In Advances in Neural Information Processing Systems, Vol. 37, NeurIPS 2024.
> 7. Brukhim, N., Daniely, A., Mansour, Y., & Moran, S. Multiclass Boosting: Simple and Intuitive Weak Learning Criteria. In Advances in Neural Information Processing Systems, Vol. 36, NeurIPS 2023.
> 8. Hanneke, S., Raman, V., Shaeiri, A., & Subedi, U. Multiclass Transductive Online Learning. In Advances in Neural Information Processing Systems, Vol. 37, NeurIPS 2024.
> 9. Hanneke, S., Moran, S., & Zhang, Q. Improved Sample Complexity for Multiclass PAC Learning. In Advances in Neural Information Processing Systems, Vol. 37, NeurIPS 2024.
> 10. Raman, V., Subedi, U., & Tewari, A. Smoothed Online Classification can be Harder than Batch Classification. In Advances in Neural Information Processing Systems, Vol. 37, NeurIPS 2024.
> 11. Kalavasis, A., Mehrotra, A., & Velegkas, G. On the limits of language generation: Trade-offs between hallucination and mode-collapse. In Proceedings of the 57th Annual ACM Symposium on Theory of Computing, STOC 2025.

---

> > ### Comment · Reviewer_wMkv · 2025-08-05
> >
> > Thanks. You point is well received. I have increased my score.

---

> > > ### Author Response · Authors · 2025-08-05
> > >
> > > We thank the reviewer for dedicating their time to reassess our work and appreciate their feedback.

---

### Official Review · Reviewer_Y85r · 2025-06-29

**Clarity:** 3
**Significance:** 3
**Originality:** 3
**Rating:** 4
**Confidence:** 4

**Summary:**

The authors consider multiclass classification in the non-uniform learning framework. They establish structural results showing that in contrast to the uniform and universal learning framework, the non-uniform learning framework exhibits a separation between learning under full supervision and learning with bandit feedback; that is, when the learner only observes whether or not the prediction is correct. In more detail, the authors characterize the hypothesis classes which are non-uniformly learnable under bandit feedback in both the online and PAC settings by presenting them as countable unions of classes with bounded dimensions which are known to characterize learnability in the corresponding uniform framework. The authors also exhibit a strict separation between bandit feedback and full supervision in the non-uniform framework by exhibiting a specific hypothesis class which is learnable under full supervision (even in the uniform learning framework) but not learnable with bandit feedback in the non-uniform framework.

**Questions:**

I would appreciate it if the authors could elaborate on the technical novelties of their results, specifically, what additional techniques are needed which are not present in more classical results relating uniform and non-uniform learnability. As it stands, it seems to me like a quite general statement could be proven: A class $H$ is non-uniformly learnable in the X setting if and only if $H$ is a countable union of classes which are uniformly learnable in the X setting (in which case there is most likely a known dimension characterizing such classes). Is my intuition correct? Do the authors believe such a general statement could be proven formally?

**Ethical Concerns:**

["NO or VERY MINOR ethics concerns only"]

**Final Justification:**

It seems that the major technical contribution of this paper was to "construct a hypothesis class that is non-uniformly learnable under full supervision in the adversarial online model (and thus also in the i.i.d. batch model), but not non-uniformly learnable under bandit feedback in the i.i.d. batch model". While I believe this result is interesting, the rest of the results presented in the paper seem to follow from standard arguments from the learning theory literature and are not very technically interesting. I am still leaning towards acceptance, though I do feel like the technical contribution are a bit lacking. I therefore maintain my overall score and I will not champion this paper.

**Limitations:**

yes

**Quality:**

3

**Strengths And Weaknesses:**

Strengths:

* This work proves several open questions in the literature of learnability under bandit feedback, of which previous works considered only the uniform and the universal learning frameworks. As the non-uniform learning framework is well-studied in the learning theory literature, studying the role of bandit feedback in this framework seems natural.
* The results are presented clearly and are easy to understand.
* The fact that in the non-uniform learning framework there is a strict separation between full supervision and bandit feedback is surprising given that such a separation does not exist in the uniform and the universal learning frameworks.

Weaknesses:

* From a technical perspective, the results established in this work are obtained in a fairly straightforward manner from existing results in the learning theory literature. For example, showing the equivalence of non-uniform learnability under bandit feedback with the structural property of being a countable union of classes with finite BL-dim seems quite analogous to the corresponding result from binary classification - where a hypothesis class is non-uniformly learnable if and only if it is a countable union of classes with finite VC dimension. The structural properties presented in this paper seem to follow very similar concepts.

---

> ### Author Rebuttal · Authors · 2025-07-31
>
> We thank the reviewer for dedicating their time to assess our work. We are delighted that the reviewer mentioned that our work answers several natural open questions in the literature, found our results clear and easy to understand, and moreover, highlighted that our separation is surprising. Below, we address comments provided by the reviewer.
>
>
> > "From a technical perspective, the results established in this work are obtained in a fairly straightforward manner from existing results in the learning theory literature. For example, showing the equivalence of non-uniform learnability under bandit feedback with the structural property of being a countable union of classes with finite BL-dim seems quite analogous to the corresponding result from binary classification - where a hypothesis class is non-uniformly learnable if and only if it is a countable union of classes with finite VC dimension. The structural properties presented in this paper seem to follow very similar concepts."
> > "I would appreciate it if the authors could elaborate on the technical novelties of their results, specifically, what additional techniques are needed which are not present in more classical results relating uniform and non-uniform learnability."
>
> We construct a hypothesis class that is non-uniformly learnable under full supervision in the adversarial online model (and thus also in the i.i.d. batch model), but not non-uniformly learnable under bandit feedback in the i.i.d. batch model (and thus also not in the adversarial online model). We strongly believe this construction is novel, as the hypothesis class differs significantly from natural or commonly considered classes. In summary, this serves as our main novel technical contribution that reveals a fundamental distinction between the non-uniform and universal learning frameworks.
>
>
> > "As it stands, it seems to me like a quite general statement could be proven: A class  is non-uniformly learnable in the X setting if and only if  is a countable union of classes which are uniformly learnable in the X setting (in which case there is most likely a known dimension characterizing such classes). Is my intuition correct? Do the authors believe such a general statement could be proven formally?"
>
> Yes, we share the same intuition. Moreover, we believe that such a general statement can indeed be formally proven. However, we found it challenging to define a sufficiently general learning framework within which this result can be elegantly established.
>
>
> Finally, once again, we thank the reviewer for dedicating their time to assess our work. We hope this rebuttal convinces you of the novelty of our contribution.

---

> > ### Comment · Reviewer_Y85r · 2025-08-03
> >
> > I thank the authors for their response.
> >
> > I now better understand that the main technical contribution of this paper is in the construction of a hypothesis class which shows a separation between full supervision and bandit feedback in the non-uniform setting, and not necessarily the characterization of bandit non-uniform learnability by itself. I think the authors should be more explicit in the final version and make clear which technical result rely on more standard techniques and which techniques are more novel.

---

> ### Author Response · Authors · 2025-08-05
>
> We thank the reviewer for dedicating their time to reassess our work.
>
> We appreciate the reviewer's feedback. We completely agree with the reviewer's comment. Following your suggestion, we will make sure to clarify which results rely on more standard techniques and which rely on novel techniques for the camera-ready version.
>
> Once again, thank you for dedicating your time to reassessing our work.

---

### Official Review · Reviewer_fKQT · 2025-07-02

**Clarity:** 2
**Significance:** 3
**Originality:** 3
**Rating:** 4
**Confidence:** 3

**Summary:**

The paper studies learnability under uniform, non-uniform and universal learning frameworks, as recently investigated by Hanneke et al 2025a,b. Among the contributions: (i) the paper gives a combinatorial characterization in the non-uniform learning framework, and (ii) the authors are able to construct a hypothesis class that is non-uniformly learnable under full supervision in the adversarial online model but not non-uniformly learnable under bandit feedback, revealing a separation between the non-uniform and universal learning settings (in the universal setting we have an exact equivalences between multiclass learnability under bandit feedback and full feedback).

**Questions:**

- What is the universal learning setting ?
- What is the effective label space ?
- Please articulate the applicability of your setting to real-world problems. E.g., why is it more appealing to rely on an algorithm that also works on infinite label spaces when the label space is finite but large, if we only have bandit feedback available.

**Ethical Concerns:**

["NO or VERY MINOR ethics concerns only"]

**Final Justification:**

Reasonable but perhaps not groundbreaking contribution on a line of research that is somewhat a niche. The paper should be made more accessible through an improved presentation to a wider Neurips audience.

**Limitations:**

See above.

**Quality:**

3

**Strengths And Weaknesses:**

On the positive side, I praise the authors' efforts to try and explain the frameworks and the scope of their results.

On the negative side, I'm not sure how compelling or widely interesting such results are (in general), and in particular for the Neurips audience. In this respect, I am not buying the authors' motivations in L. 271 of the paper: "in multiclass settings, it is preferable for guarantees to remain independent of the number of labels, even when finite. [...] mathematical frameworks involving infinity mostly provide clearer insights." I don't see practical motivations behind such a line of research nor do the authors attempt to provide any.

Presentation-wise, the paper leaves a lot to be desired. It is a bit awkward that, up until page 6, there is not a clear explanation of the specifics of the learning frameworks the authors are alluding to. Though I am very much aware of the basics, I am not familiar with the difference between non-uniform and universal, and I don't see any clear explanation of that in the paper, despite the lengthy (and quite redundant) introductory sections. Moreover, as I go over Sect. 4 (e.g., Lemma 4.1) I stumble into undefined concepts (like ``effective label space"), that the authors seem to take for granted.

Minor points:
-  l. 58, "addressing" --> "address" ?
- L. 106 "unless": is it "until" rather than "unless" ?
- L. 150 "bandit feedback setting's feedback": how about just "bandit feedback" ?
- L. 159-160: upper bound and lower bound: which upper and lower bound, exactly ?
- L. 164 "bounded littlestone dimension": this comes out of the blue, frankly, at this point ...
- L. 199: what is C ?
- L. 203 "finite effective label space": same issue, it comes out of the blue in the flow of the presentation
- L. 307: I assume the history also contains $X_t$ ?
- L. 341 "we need the following a result" (typo). Btw, what is ``the following result" you are alluding to here ? Is it Thm B.1 from Daniely et al. 2011 ? If so, you cannot write things this way... you should recall the content of the theorem on the spot.

---

> ### Author Rebuttal · Authors · 2025-07-31
>
> We thank the reviewer for dedicating their time to assess our work. We are delighted that the reviewer mentioned that our work involves clear explanations. We will make sure to correct typos and incorporate minor suggestions for the camera-ready version. Below, we address comments provided by the reviewer.
>
>
> > "On the negative side, I'm not sure how compelling or widely interesting such results are (in general), and in particular for the Neurips audience. In this respect, I am not buying the authors' motivations in L. 271 of the paper: "in multiclass settings, it is preferable for guarantees to remain independent of the number of labels, even when finite. [...] mathematical frameworks involving infinity mostly provide clearer insights." I don't see practical motivations behind such a line of research nor do the authors attempt to provide any."
>
> > "Please articulate the applicability of your setting to real-world problems. E.g., why is it more appealing to rely on an algorithm that also works on infinite label spaces when the label space is finite but large, if we only have bandit feedback available."
>
>
> In practice, tasks such as image object recognition and next-word prediction involve a large number of classes. Therefore, it is desirable for theoretical guarantees to avoid dependence on the number of labels, which can be very large. For more details, we shall suggest that the reviewer see, for example, [1].
>
>
> > "Presentation-wise, the paper leaves a lot to be desired. It is a bit awkward that, up until page 6, there is not a clear explanation of the specifics of the learning frameworks the authors are alluding to. Though I am very much aware of the basics, I am not familiar with the difference between non-uniform and universal, and I don't see any clear explanation of that in the paper, despite the lengthy (and quite redundant) introductory sections. Moreover, as I go over Sect. 4 (e.g., Lemma 4.1) I stumble into undefined concepts (like ``effective label space"), that the authors seem to take for granted."
>
> We appreciate the reviewer's feedback. We completely agree with the reviewer's comments. Following your suggestion, we will make sure to incorporate relevant changes for the camera-ready version.
>
>
> > "What is the universal learning setting ?"
>
>
> The "universal" terminology of [2] refers to the "$\forall$" universal quantifier: for all realizable distributions, the error converges at some rate (though perhaps not uniformly so over distributions); analogously, in the adversarial online setting we are interested in whether, for all realizable data sequences, the number of mistakes is finite (though perhaps not uniformly so over data sequences).
>
>
> > "What is the effective label space ?"
>
> The size of the effective label space is defined as follows: $\sup_{x\in\mathcal{X}} |\{h(x):h\in\mathcal{H}\}|$.
>
>
> Finally, once again, we thank the reviewer for dedicating their time to assess our work. We hope this rebuttal addresses the points that you mentioned.
>
>
> [1] Brukhim, N., Carmon, D., Dinur, I., Moran, S., & Yehudayoff, A. (2022). A Characterization of Multiclass Learnability. In Proceedings of the 2022 IEEE 63rd Annual Symposium on Foundations of Computer Science (FOCS), pp. 943–955.
>
> [2] O. Bousquet, S. Hanneke, S. Moran, R. van Handel, and A. Yehudayoff. A Theory of Universal Learning. In Proceedings of the 53rd Annual ACM Symposium on Theory of Computing (STOC), 2021.

---

> > ### Comment · Reviewer_fKQT · 2025-08-03
> > **Response to the authors' rebuttal**
> >
> > Thanks for your response. Yet, I'm not sure how much interest this paper would generate within such a wide audience as Neurips. So, I stick to my original assessment on this one.

---

> ### Author Response · Authors · 2025-08-05
>
> We thank the reviewer for dedicating their time to reassess our work. Below, we address the new comment provided by the reviewer.
>
> The interest in multiclass learning when the number of labels can be unbounded is driven by several motivations. Firstly, guarantees for the multiclass setting should not inherently depend on the number of labels, even when it is finite. For instance, imagine we have a generalization bound that depends on the number of labels. As a result, when the number of labels is very large, the bound is vacuous, say, even if we have one million samples. However, it could be the case that the problem is learnable with only thousands of samples. Therefore, we are interested in theories that can capture the real hardness of the problem. Secondly, in mathematics, concepts involving infinities often provide cleaner insights. Thirdly, insights from this problem might also advance understanding of real-valued regression problems [1]. Finally, on a practical front, many crucial machine learning tasks involve classification into extremely large label spaces. For instance, in image object recognition, the number of classes corresponds to the variety of recognizable objects. As another example, in next-word prediction, the class count expands with the dictionary size. For more details, we shall suggest that the reviewer see, for example, [2].
>
> We respectfully note that all of the related subjects to our paper, including multiclass learning with unbounded label space, non-uniform learning, and contextual bandits, have been the focus of many papers in the recent history of NeurIPS. Indeed, they have also been the subject of considerable attention in the learning theory community, especially in recent years. For instance, there are several papers on unbounded label space, including [3, 4, 5, 6], which appeared at NeurIPS 2023, 2024 (spotlight), 2024, and 2024, respectively. In addition, Lu studied non-uniform online binary classification, presenting it as a spotlight talk at NeurIPS 2024 [7]. In close relation to the non-uniform learning framework, the following works on the universal learning framework [8, 9, 10, 11, 12] appeared at NeurIPS 2022 (oral), 2022, 2024, 2024, and 2024, respectively. Finally, we note that the notion of non-uniform learning has also recently appeared in the context of language generation theory.
>
> We hope this rebuttal convinces you further of the relevance of our contributions to the NeurIPS community.
>
> Once again, thank you for dedicating your time to reassessing our work.
>
>
> 1. Attias, I., Hanneke, S., Kalavasis, A., Karbasi, A., & Velegkas, G. Optimal Learners for Realizable Regression: PAC Learning and Online Learning. In Advances in Neural Information Processing Systems, Vol. 36, NeurIPS 2023.
> 2. Brukhim, N., Carmon, D., Dinur, I., Moran, S., & Yehudayoff, A. A Characterization of Multiclass Learnability. In Proceedings of the 2022 IEEE 63rd Annual Symposium on Foundations of Computer Science, FOCS 2022.
> 3. Brukhim, N., Daniely, A., Mansour, Y., & Moran, S. Multiclass Boosting: Simple and Intuitive Weak Learning Criteria. In Advances in Neural Information Processing Systems, Vol. 36, NeurIPS 2023.
> 4. Hanneke, S., Raman, V., Shaeiri, A., & Subedi, U. Multiclass Transductive Online Learning. In Advances in Neural Information Processing Systems, Vol. 37, NeurIPS 2024.
> 5. Hanneke, S., Moran, S., & Zhang, Q. Improved Sample Complexity for Multiclass PAC Learning. In Advances in Neural Information Processing Systems, Vol. 37, NeurIPS 2024.
> 6.	Raman, V., Subedi, U., & Tewari, A. Smoothed Online Classification can be Harder than Batch Classification. In Advances in Neural Information Processing Systems, Vol. 37, NeurIPS 2024.
> 7. Lu, Z. When Is Inductive Inference Possible? In Advances in Neural Information Processing Systems, Vol. 37, NeurIPS 2024.
> 8. Hanneke, S., Karbasi, A., Moran, S., & Velegkas, G. Universal Rates for Interactive Learning. In Advances in Neural Information Processing Systems, Vol. 35, NeurIPS 2022.
> 9. Kalavasis, A., Velegkas, G., & Karbasi, A. Multiclass Learnability Beyond the PAC Framework: Universal Rates and Partial Concept Classes. In Advances in Neural Information Processing Systems, Vol. 35, NeurIPS 2022.
> 10. Hanneke, S., Karbasi, A., Moran, S., & Velegkas, G. Universal Rates for Active Learning. In Advances in Neural Information Processing Systems, Vol. 37, NeurIPS 2024.
> 11. Hanneke, S., & Xu, M. Universal Rates of Empirical Risk Minimization. In Advances in Neural Information Processing Systems, Vol. 37, NeurIPS 2024.
> 12. Hanneke, S., & Wang, H. A Theory of Optimistically Universal Online Learnability for General Concept Classes. In Advances in Neural Information Processing Systems, Vol. 37, NeurIPS 2024.

---

> > ### Comment · Reviewer_fKQT · 2025-08-05
> > **Response to authors**
> >
> > Thanks to the authors for bringing to my attention that a number of papers in the infinite multi-class setting have already appeared in Neurips. In light of that, and under the assumption that the authors' will be able to make the paper more accessible by an improved presentation, I am raising my score.

---

> > > ### Author Response · Authors · 2025-08-06
> > >
> > > We thank the reviewer for dedicating their time to reassess our work and appreciate their feedback. We will ensure that your suggestion is incorporated into the camera-ready version.

---

### Official Review · Reviewer_fkCf · 2025-07-03

**Clarity:** 3
**Significance:** 4
**Originality:** 3
**Rating:** 4
**Confidence:** 3

**Summary:**

The paper studies when multiclass hypothesis classes remain learnable from bandit-only feedback in the “non-uniform” setting that sits between classic PAC learning and universal learning. Learnability in the i.i.d. (PAC) regime occurs iff the class can be decomposed into countably many subclasses, each with finite effective label space and finite Natarajan dimension. In the adversarial online regime, learnability occurs iff the class is a countable union of subclasses with finite bandit Littlestone dimension. To match these information-theoretic conditions, the authors present a generic Bandit-SOA algorithm as well as separation examples, showing the non-uniform and universal frameworks fundamentally diverge.

**Questions:**

Are the non-uniform learners implementable in polynomial time for finite-label classes with small BLdim/Natarajan dim? If not, is there any hope of an efficient surrogate?

**Ethical Concerns:**

["NO or VERY MINOR ethics concerns only"]

**Final Justification:**

I am still not quite sure about whether the novelty of the hypothesis class construction for separating full information and bandit setting has enough broader relevance. Aside from that, I am OK with the paper.

**Quality:**

4

**Strengths And Weaknesses:**

Strength:
1. The paper proves tight if-and-only-if characterizations for non-uniform multiclass learning under bandit feedback in both PAC and online settings (Theorems 1.1 - 1.3). Proofs are mostly self-contained and appear to build correctly on previous lines of work.
2. Resolves an open question left by Hanneke et al. (2025). The negative separation example (Prop 1.4) shows a fundamental distinction between bandit learning and full supervised learning in the non-uniform framework.
3. Extends non-uniform learnability from binary to multiclass and bandit feedback, which (to the best of my knowledge) has not been done; the combinatorial conditions seem new.

Weakness:
1. Practical relevance is indirect: all algorithms rely on full knowledge of countable decompositions and invoke BSOA sub-routines with exponential overhead. The paper does not discuss computational complexity or whether efficient learners exist even for finite-label subclasses.
2. It seems the generic “countable union + mistake/sample budget allocator” technique is adapted from Benedek and Itai (1988); novelty mainly lies in multiclass extension. Would like to see more clarity on what new proof ideas are required beyond existing reductions.

---

> ### Author Rebuttal · Authors · 2025-07-31
>
> We thank the reviewer for dedicating their time to assess our work. We are delighted that the reviewer mentioned that our work involves a fundamental distinction and new results, and moreover, found our proofs self-contained and sound. Below, we address comments provided by the reviewer.
>
>
> > "Practical relevance is indirect: all algorithms rely on full knowledge of countable decompositions and invoke BSOA sub-routines with exponential overhead. The paper does not discuss computational complexity or whether efficient learners exist even for finite-label subclasses."
>
> > "Are the non-uniform learners implementable in polynomial time for finite-label classes with small BLdim/Natarajan dim? If not, is there any hope of an efficient surrogate?"
>
> We appreciate the reviewer's feedback. We completely agree with the reviewer's comment. Following your suggestion, we will make sure to add such a discussion for the camera-ready version.
>
> Regarding the computational complexity of our algorithms, we note that our algorithms are not efficient (not polynomial time) for general concept classes. However, this is an issue for both PAC and adversarial online learning even in the binary setting. For instance, in the case of adversarial online learning, SOA involves computing the Littlestone dimension of concept classes defined by the online learner in the course of its interaction with the adversary, which is a challenging computation, even when the concept class and the set of features are finite [1]. Notably, no efficient algorithm can achieve finite mistake bounds for general Littlestone classes [2].
>
>
> > "It seems the generic "countable union + mistake/sample budget allocator" technique is adapted from Benedek and Itai (1988); novelty mainly lies in multiclass extension. Would like to see more clarity on what new proof ideas are required beyond existing reductions."
>
> We construct a hypothesis class that is non-uniformly learnable under full supervision in the adversarial online model (and thus also in the i.i.d. batch model), but not non-uniformly learnable under bandit feedback in the i.i.d. batch model (and thus also not in the adversarial online model). We strongly believe this construction is novel, as the hypothesis class differs significantly from natural or commonly considered classes. In summary, this serves as our main novel technical contribution that reveals a fundamental distinction between the non-uniform and universal learning frameworks.
>
>
> Finally, once again, we thank the reviewer for dedicating their time to assess our work. We hope this rebuttal convinces you of the novelty of our contribution and addresses your question.
>
>
> [1] P. Manurangsi and A. Rubinstein. Inapproximability of VC Dimension and Littlestone’s Dimension. In Proceedings of the 30th Conference on Learning Theory (COLT), 2017.
>
> [2] A. Assos, I. Attias, Y. Dagan, C. Daskalakis, and M. K. Fishelson. Online Learning and Solving Infinite Games with an ERM Oracle. In Proceedings of the 36th Conference on Learning Theory (COLT), 2023.

---

> > ### Comment · Reviewer_fkCf · 2025-08-06
> >
> > Thank you for the response. I will maintain my score.

---

### Official Review · Reviewer_sVPb · 2025-07-04

**Clarity:** 3
**Significance:** 3
**Originality:** 3
**Rating:** 5
**Confidence:** 2

**Summary:**

The paper studies multiclass prediction when only bandit feedback is available, and the number of possible labels may be infinite. Under the uniform framework no class is learnable once labels are unbounded, and prior work show that under the universal framework bandit feedback is as powerful as full supervision. This work studies the non-uniform setting, and relate to learnability to Natarajan dimension in the PAC setting and bandit Littlestone dimension in the online learning setting.

**Questions:**

The algorithm based on BSOA is used to learn non-uniformly under bandit feedback in the online setting. How does the algorithm perform in terms of computational complexity?

Can the results be generalized to other forms of feedback?

**Ethical Concerns:**

["NO or VERY MINOR ethics concerns only"]

**Final Justification:**

The results the paper presented are interesting and new (in the sense that no one has claimed them before), but technically not very surprising. Some revision is needed to make it more accessible. I keep my score to an accept.

**Limitations:**

yes

**Quality:**

3

**Strengths And Weaknesses:**

As someone not closely following this line of work, the result in this paper seemed interesting and substantial to me. In particular, they give a nice exact characterization of the learnability in the non-uniform setting. The theoretical results seem very clean.

While the results are new, the proofs are relatively simple, and does not seem to lead to new techniques or insights. Hence, I am not quite sure how surprising the results should be considered. To me, the results feel more or less direct analogs to some of the classical results in non-uniform learnability, for example, Chapter 7 in the book [1]. As such, I think some of the classical results in non-uniform learning should be mentioned somewhere, and a comparison should be made.

The paper is structured oddly in places. For example, it seems weird that the definition for various complexity measures only appears in the appendix. This makes reading the paper somewhat difficult.

In the introduction, a table summarizing and juxtaposing the three frameworks and learnability criteria, and highlighting some key examples, could be helpful. A more concrete illustration and examples around the key definitions and theorems would also be helpful.

[1] Understanding machine learning, Shalev-Shwartz and Ben-David.

Other comments:
- Apperances of section X should be capitalized (Section X).
- Section 5 have just one subsection and nothing else. If only one subsection, just present as a section.

---

> ### Author Rebuttal · Authors · 2025-07-31
>
> We thank the reviewer for dedicating their time to assess our work. We are delighted that the reviewer found our results interesting, and moreover, mentioned that our proofs are clean. We will make sure to correct typos and incorporate minor suggestions for the camera-ready version. Below, we address major comments provided by the reviewer.
>
>
> > "While the results are new, the proofs are relatively simple, and does not seem to lead to new techniques or insights. Hence, I am not quite sure how surprising the results should be considered. To me, the results feel more or less direct analogs to some of the classical results in non-uniform learnability, for example, Chapter 7 in the book [1]. As such, I think some of the classical results in non-uniform learning should be mentioned somewhere, and a comparison should be made."
>
> We construct a hypothesis class that is non-uniformly learnable under full supervision in the adversarial online model (and thus also in the i.i.d. batch model), but not non-uniformly learnable under bandit feedback in the i.i.d. batch model (and thus also not in the adversarial online model). We strongly believe this construction is novel, as the hypothesis class differs significantly from natural or commonly considered classes. In summary, this serves as our main novel technical contribution that reveals a fundamental distinction between the non-uniform and universal learning frameworks.
>
> We appreciate the reviewer's feedback. We completely agree with the reviewer's comment on the comparison. Following your suggestion, we will make sure to add such a discussion for the camera-ready version.
>
>
> > "The paper is structured oddly in places. For example, it seems weird that the definition for various complexity measures only appears in the appendix. This makes reading the paper somewhat difficult."
>
> We appreciate the reviewer's feedback. We completely agree with the reviewer's comment on the paper structure. Following your suggestion, we will make sure to change the structure of the paper for the camera-ready version.
>
>
> > "In the introduction, a table summarizing and juxtaposing the three frameworks and learnability criteria, and highlighting some key examples, could be helpful."
>
> We appreciate the reviewer's feedback. We completely agree with the reviewer's comment about the table. Following your suggestion, we will make sure to include a table for the camera-ready version.
>
>
> > "The algorithm based on BSOA is used to learn non-uniformly under bandit feedback in the online setting. How does the algorithm perform in terms of computational complexity?"
>
> Regarding the computational complexity of our algorithms, we note that our algorithms are not efficient for general concept classes. However, this is an issue for both PAC and adversarial online learning even in the binary setting. For instance, in the case of adversarial online learning, SOA involves computing the Littlestone dimension of concept classes defined by the online learner in the course of its interaction with the adversary, which is a challenging computation, even when the concept class and the set of features are finite [1]. Notably, no efficient algorithm can achieve finite mistake bounds for general Littlestone classes [2].
>
>
> > "Can the results be generalized to other forms of feedback?"
>
> We appreciate the reviewer's feedback. It is a very interesting question whether these results can be generalized to other forms of feedback. We believe this is a meaningful question and have listed it as one of the future directions.
>
>
> Finally, once again, we thank the reviewer for dedicating their time to assess our work. We hope this rebuttal addresses the points that you mentioned.
>
>
> [1] P. Manurangsi and A. Rubinstein. Inapproximability of VC Dimension and Littlestone’s Dimension. In Proceedings of the 30th Conference on Learning Theory (COLT), 2017.
>
> [2] A. Assos, I. Attias, Y. Dagan, C. Daskalakis, and M. K. Fishelson. Online Learning and Solving Infinite Games with an ERM Oracle. In Proceedings of the 36th Conference on Learning Theory (COLT), 2023.

---

> > ### Comment · Reviewer_sVPb · 2025-08-06
> >
> > Thank you for the response. I will keep my score.

---

### Decision · Program_Chairs · 2025-09-17

**Decision:**

Accept (poster)

**Comment:**

The authors study the problem of multiclass non-uniform learning with bandit or full supervision in i.i.d. batch and adversarial online models. A key contribution is constructing a hypothesis class that is non-uniformly learnable under full supervision in the adversarial online model (and thus also in the i.i.d. batch model) but not non-uniformly learnable under bandit feedback in the i.i.d. batch model. This separation is an interesting result on a well defined problem. The problem is rather niche (within the theory community) and it is not clear if the proofs have take home messages for other problems. Nevertheless, the paper offers a nice resolution to a well defined problem and therefore I recommend acceptance.